# The potassium channel subunit $K_V$1.8 (*Kcna10*) is essential for the distinctive outwardly rectifying conductances of type I and II vestibular hair cells

Hannah R Martin[1], Anna Lysakowski[2], Ruth Anne Eatock[1]*

[1]Department of Neurobiology, University of Chicago, Chicago, United States; [2]Department of Anatomy and Cell Biology, University of Illinois at Chicago, Chicago, United States

## eLife assessment

This study provides direct evidence showing that $K_V$1.8 channels provide the basis for several potassium currents in the two types of sensory hair cells found in the mouse vestibular system. This is an **important** finding because the nature of the channels underpinning the unusual potassium conductance $g_{K,L}$ in type I hair cells has been under scrutiny for many years. The experimental evidence is **compelling** and the analysis is rigorous. The study will be of interest to cell and molecular biologists as well as vestibular and auditory neuroscientists.

*For correspondence: eatock@uchicago.edu

**Abstract** In amniotes, head motions and tilt are detected by two types of vestibular hair cells (HCs) with strikingly different morphology and physiology. Mature type I HCs express a large and very unusual potassium conductance, $g_{K,L}$, which activates negative to resting potential, confers very negative resting potentials and low input resistances, and enhances an unusual non-quantal transmission from type I cells onto their calyceal afferent terminals. Following clues pointing to $K_V$1.8 (*Kcna10*) in the Shaker K channel family as a candidate $g_{K,L}$ subunit, we compared whole-cell voltage-dependent currents from utricular HCs of $K_V$1.8-null mice and littermate controls. We found that $K_V$1.8 is necessary not just for $g_{K,L}$ but also for fast-inactivating and delayed rectifier currents in type II HCs, which activate positive to resting potential. The distinct properties of the three $K_V$1.8-dependent conductances may reflect different mixing with other $K_V$ subunits that are reported to be differentially expressed in type I and II HCs. In $K_V$1.8-null HCs of both types, residual outwardly rectifying conductances include $K_V$7 (*Knq*) channels. Current clamp records show that in both HC types, $K_V$1.8-dependent conductances increase the speed and damping of voltage responses. Features that speed up vestibular receptor potentials and non-quantal afferent transmission may have helped stabilize locomotion as tetrapods moved from water to land.

## Introduction

The receptor potentials of hair cells (HCs) are strongly shaped by large outwardly rectifying K+ conductances that are differentially expressed according to HC type. Here, we report that a specific voltage-gated K+ ($K_V$) channel subunit participates in very different $K_V$ channels dominating the membrane conductances of type I and II HCs in amniote vestibular organs.

Type I HCs occur only in amniote vestibular organs. Their most distinctive features are that they are enveloped by a calyceal afferent terminal (*Wersall, 1956*; *Lysakowski and Goldberg, 2004*) and that

they express $g_{K,L}$ (*Correia and Lang, 1990*; *Rennie and Correia, 1994*; *Rüsch and Eatock, 1996a*): a large non-inactivating conductance with an activation range from –100 to –60 mV, far more negative than other 'low-voltage-activated' $K_V$ channels. HCs are known for their large outwardly rectifying $K^+$ conductances, which repolarize membrane voltage following a mechanically evoked perturbation and in some cases contribute to sharp electrical tuning of the HC membrane. $g_{K,L}$ is unusually large and unusually negatively activated, and therefore strongly attenuates and speeds up the receptor potentials of type I HCs (*Correia et al., 1996*; *Rüsch and Eatock, 1996b*). In addition, $g_{K,L}$ augments non-quantal transmission from type I HC to afferent calyx by providing open channels for $K^+$ flow into the synaptic cleft (*Contini et al., 2012*; *Contini et al., 2017*; *Contini et al., 2020*; *Govindaraju et al., 2023*), increasing the speed and linearity of the transmitted signal (*Songer and Eatock, 2013*).

Type II HCs have compact afferent synaptic contacts (boutons) where the receptor potential drives quantal release of glutamate. They have fast-inactivating (A-type, $g_A$) and delayed rectifier ($g_{DR}$) conductances that are opened by depolarization above resting potential ($V_{rest}$).

The unusual properties of $g_{K,L}$ have long attracted curiosity about its molecular nature. $g_{K,L}$ has been proposed to include 'M-like' $K_V$ channels in the $K_V7$ and/or erg channel families (*Kharkovets et al., 2000*; *Hurley et al., 2006*; *Holt et al., 2007*). The $K_V7.4$ subunit was of particular interest because it contributes to the low-voltage-activated conductance, $g_{K,n}$, in cochlear outer HCs, but was eventually eliminated as a $g_{K,L}$ subunit by experiments on $K_V7.4$-null mice (*Spitzmaul et al., 2013*).

Several observations suggested the $K_V1.8$ (KCNA10) subunit as an alternative candidate for $g_{K,L}$. $K_V1.8$ is highly expressed in vestibular sensory epithelia (*Carlisle et al., 2012*), particularly HCs (*Lee et al., 2013*; *Scheffer et al., 2015*; *McInturff et al., 2018*), with slight expression elsewhere (skeletal muscle, *Lee et al., 2013*; kidney, *Yao et al., 2002*). *Kcna10*$^{-/-}$ mice show absent or delayed vestibular-evoked potentials, the synchronized activity of afferent nerve fibers sensitive to fast linear head motions (*Lee et al., 2013*). Unique among $K_V1$ channels, $K_V1.8$ has a cyclic nucleotide-binding domain (*Lang et al., 2000*) with the potential to explain $g_{K,L}$'s known cGMP dependence (*Behrend et al., 1997*; *Chen and Eatock, 2000*).

Our comparison of whole-cell currents and immunohistochemistry in type I HCs from *Kcna10*$^{-/-}$ and *Kcna10*$^{+/+,+/-}$ mouse utricles showed that $K_V1.8$ expression is necessary for $g_{K,L}$. More surprisingly, $K_V1.8$ expression is also required for A-type and delayed rectifier conductances of type II HCs. In both HC types, eliminating the $K_V1.8$-dependent major conductances revealed a smaller delayed rectifier conductance involving $K_V7$ channels. Thus, the distinctive outward rectifiers that produce such different receptor potentials in type I and II HCs both include $K_V1.8$ and $K_V7$ channels.

## Results

We compared whole-cell voltage-activated $K^+$ currents in type I and II HCs from homozygous knockout (*Kcna10*$^{-/-}$) animals and their wildtype (*Kcna10*$^{+/+}$) or heterozygote (*Kcna10*$^{+/-}$) littermates. We immunolocalized $K_V1.8$ subunits in the utricular epithelium and pharmacologically characterized the residual $K^+$ currents of *Kcna10*$^{-/-}$ animals. Current clamp experiments demonstrated the impact of $K_V1.8$-dependent currents on passive membrane properties.

We recorded from three utricular zones: lateral extrastriola (LES), striola, and medial extrastriola (MES); striolar and extrastriolar zones have many structural and functional differences and give rise to afferents with different physiology (reviewed in *Goldberg, 2000*; *Eatock and Songer, 2011*). Recordings are from 412 type I and II HCs (53% LES, 30% MES, 17% striola) from mice between postnatal day (P) 5 and P370. We recorded from such a wide age range to test for developmental or senescent changes in the impact of the null mutation. Above P18, we did not see substantial changes in $K_V$ channel properties, as reported (*González-Garrido et al., 2021*).

*Kcna10*$^{-/-}$ animals appeared to be healthy and to develop and age normally, as reported (*Lee et al., 2013*), and HCs were healthy (see Methods for criteria).

### $K_V1.8$ is necessary for $g_{K,L}$ in type I HCs

The large low-voltage-activated conductance, $g_{K,L}$, in *Kcna10*$^{+/+,+/-}$ type I HCs produces distinctive whole-cell current responses to voltage steps, as highlighted by our standard type I voltage protocol (*Figure 1A*). From a holding potential within the $g_{K,L}$ activation range (here –74 mV), the cell is hyperpolarized to –124 mV, negative to $E_K$ and the activation range, producing a large inward current

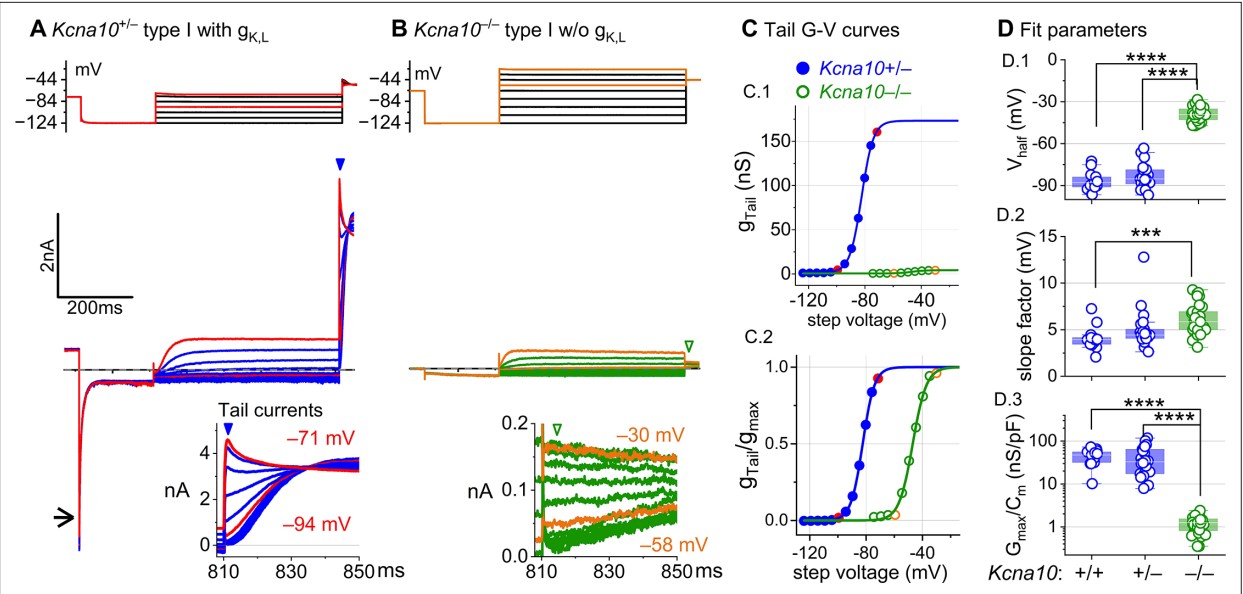

**Figure 1.** $Kcna10^{-/-}$ type I hair cells (HCs) lacked $g_{K,L}$, the dominant conductance in mature $Kcna10^{+/+,++/-}$ type I HCs. Representative voltage-evoked currents in (**A**) a P22 $Kcna10^{+/-}$ type I HC and (**B**) a P29 $Kcna10^{-/-}$ type I HC. (**A**) *Arrow,* transient inward current that is a hallmark of $g_{K,L}$. *Arrowheads,* tail currents, magnified in *insets.* For steps positive to the midpoint voltage, tail currents are very large. As a result, K+ accumulation in the calyceal cleft reduces driving force on K+, causing currents to decay rapidly, as seen in A (***Lim et al., 2011***). Note that the voltage protocol (top) in B extends to more positive voltages. (**C**) Activation (*G–V*) curves from tail currents in A and B; symbols, data; curves, Boltzmann fits (***Equation 1***). (**D**) Fit parameters from mice >P12 show big effect of $Kcna10^{-/-}$ and no difference between $Kcna10^{+/-}$ and $Kcna10^{+/+}$. (**D.1**), Tukey's test: +/+ vs –/–, p<1E-9; +/– vs –/–, p<1E-9. (**D.2**), Tukey's test: +/+ vs –/–, p=9.4E-4. (**D.3**), Tukey's test: +/+ vs –/–, p<1E-9; +/– vs –/–, p<1E-9. *Asterisks*: ***p < 0.001; and ****p < 0.0001. *Line,* median; *Box,* interquartile range; *Whiskers,* outliers. See ***Table 1*** for statistics.

The online version of this article includes the following figure supplement(s) for figure 1:

**Figure supplement 1.** Developmental changes in type I hair cell (HC) $K_V$ conductances.

through open $g_{K,L}$ channels that rapidly decays as the channels deactivate. We use the large transient inward current as a hallmark of $g_{K,L}$. The hyperpolarization closes all $g_{K,L}$ channels, and then the activation function is probed with a series of depolarizing steps, obtaining the maximum conductance from the peak tail current at –44 mV (***Figure 1A***). We detected no difference between the Boltzmann parameters of $g_{K,L}$ *G–V* curves from $Kcna10^{+/-}$ and $Kcna10^{+/+}$ type I HCs.

For a similar voltage protocol, $Kcna10^{-/-}$ type I HCs (***Figure 1B***) produced no inward transient current at the step from holding potential to –124 mV and much smaller depolarization-activated currents during the iterated steps, even at much more positive potentials. ***Figure 1C*** compares the conductance–voltage (*G–V*, activation) curves fit to tail currents (***Equation 1***; see insets in ***Figure 1A, B***): the maximal conductance ($g_{max}$) of the $Kcna10^{-/-}$ HC was over 10-fold smaller (***Figure 1C.1***), and the curve was positively shifted by >40 mV (***Figure 1C.2***). ***Figure 1D*** shows the *G–V* Boltzmann fit parameters for type I HCs from mice >P12, an age at which type I HCs normally express $g_{K,L}$ (***Rüsch et al., 1998***).

In type I HCs from $Kcna10^{+/+,+,-}$ mice, the *G–V* parameters of outwardly rectifying currents transitioned over the first 15–20 postnatal days from values for a conventional delayed rectifier, activating positive to resting potential, to $g_{K,L}$ values (***Figure 1—figure supplement 1A***), as previously described (***Rüsch et al., 1998***; ***Géléoc et al., 2004***; ***Hurley et al., 2006***). Between P5 and P10, some type I HCs have not yet acquired the physiologically defined conductance, $g_{K,L}$. No effects of $K_V$1.8 deletion were seen in the delayed rectifier currents of immature type I HCs (***Figure 1—figure supplement 1B***), showing that they were not immature forms of the $K_V$1.8-dependent $g_{K,L}$ channels.

$Kcna10^{-/-}$ type I HCs had a much smaller residual delayed rectifier that activated positive to resting potential, with $V_{half}$ ~–40 mV and $g_{max}$ density of 1.3 nS/pF. No additional K+ conductance activated up to +40 mV, and *G–V* parameters did not change much with age from P5 to P370. We characterize this $K_V$1.8-independent delayed rectifier later. A much larger non-$g_{K,L}$ delayed rectifier conductance ($g_{DR,I}$) was reported in our earlier publication on mouse utricular type I HCs (***Rüsch et al., 1998***). This current

**Table 1.** Type I hair cell $K_V$ activation voltage dependence.
Mean ± SEM (number of cells). g is effect size, Hedge's g. KWA is Kruskal–Wallis ANOVA.

| Zone | Kcna10 | Tail $V_{1/2}$, mV* | Tail $S$, mV[†] | Tail $g_{max}$, nS[‡] | Tail $g_{max}/C_m$, nS/pF[§] | Age (median, range) |
|---|---|---|---|---|---|---|
| Extrastriola | +/+ | −85 ± 2 (12) | 4.3 ± 0.4 (12) | 270 ± 40 (11) | 47 ± 8 (11) | 22, 14–287 |
| | +/− | −83 ± 1 (40) | 5.2 ± 0.3 (40) | 210 ± 20 (40) | 37 ± 4 (40) | 19, 13–259 |
| | −/− | −40.2 ± 0.9 (26) | 5.7 ± 0.3 (26) | 5.4 ± 0.3 (26) | 1.11 ± 0.08 (26) | 45, 14–277 |
| Striola | +/+ | −87 ± 3 (6) | 4.3 ± 0.3 (6) | 310 ± 70 (6) | 41 ± 7 (6) | 40, 15–59 |
| | +/− | −88 ± 2 (3) | 4.7 ± 0.9 (3) | 270 ± 60 (3) | 44 ± 6 (3) | 19, 14–20 |
| | −/− | −38 ± 1 (13) | 6.2 ± 0.4 (13) | 6.5 ± 0.6 (13) | 1.5 ± 0.1 (13) | 202, 14–370 |

*−/− vs +/+: two-way ANOVA, $p < 1E−9$, g 7.7; −/− vs +/−: two-way ANOVA, $p < 1E−9$, g 6.8.
[†]−/− vs +/+: two-way ANOVA, $p = 8.4E−4$, g 1.2.
[‡]−/− vs +/+: two-way ANOVA, $p < 1E−9$, g 3.7; −/− vs +/−: two-way ANOVA, $p < 1E−9$, g 2.1.
[§]−/− vs +/+: two-way ANOVA, $p < 1E−9$, g 3.6; −/− vs +/−: two-way ANOVA, $p < 1E−9$, g 2.0.

was identified as that remaining after 'blocking' $g_{K,L}$ with 20 mM external $Ba^{2+}$. Our new data suggest that there is no large non-$g_{K,L}$ conductance, and that instead high $Ba^{2+}$ positively shifted the apparent voltage dependence of $g_{K,L}$.

## $K_V1.8$ strongly affects type I passive properties and responses to current steps

While the cells of $Kcna10^{-/-}$ and $Kcna10^{+/-}$ epithelia appeared healthy, type I HCs had smaller membrane capacitances ($C_m$): 4–5 pF in $Kcna10^{-/-}$ type I HCs, ~20% smaller than $Kcna10^{+/-}$ type I HCs (~6 pF) and ~30% smaller than $Kcna10^{+/+}$ type I HCs (6–7 pF; *Table 2*). While $C_m$ scales with surface area, the lack of change in soma sizes by deletion of $K_V1.8$ (*Supplementary file 1b*) suggests that surface area was not different. Instead, $C$ may be higher in $Kcna10^{+/+}$ cells because of $g_{K,L}$ for two reasons. First, highly expressed trans-membrane proteins (see discussion of $g_{K,L}$ channel density in *Chen and Eatock, 2000*) can affect membrane thickness (*Mitra et al., 2004*), which is inversely proportional to specific $C_m$. Second, resistive current through $g_{K,L}$ could contaminate estimations of capacitive current, which is calculated from the decay time constant of transient current evoked by a small voltage step negative to −90 mV, where we measured $C_m$ (see Methods).

Basolateral conductances help set the resting potential and passive membrane properties that regulate the time course and gain of voltage responses to small currents. To examine the effect of

**Table 2.** Type I hair cell passive membrane properties in the extrastriola (ES) and striola (S).
Mean ± SEM (number of cells). g is effect size, Hedge's g. KWA is Kruskal–Wallis ANOVA.

| Zone | *Kcna10* | $V_{rest}$, mV*, [†] | $R_{input}$, MΩ[‡] | $\tau_{RC}$, ms[§] | $C_m$, pF[¶] | Age (median, range) |
|---|---|---|---|---|---|---|
| ES | +/+ | −84 ± 3 (6) | 44 ± 6 (6) | 0.24 ± 0.03 (6) | 6.1 ± 0.4 (13) | 20, 14–287 |
| | +/− | −88.0 ± 0.7 (28) | 55 ± 5 (24) | 0.32 ± 0.03 (23) | 5.8 ± 0.2 (44) | 21, 16–29 |
| | −/− | −63 ± 2 (15) | 1400 ± 100 (15) | 6.4 ± 0.6 (15) | 5.0 ± 0.2 (27) | 45, 14–202 |
| S | +/+ | −87 ± 2 (4) | 50 ± 8 (4) | 0.30 ± 0.04 (4) | 7.4 ± 0.7 (7) | 43, 40–59 |
| | +/− | −87 ± 3 (3) | 38 ± 8 (2) | 0.21 ± 0.01 (2) | 5.9 ± 0.6 (3) | 19, 19–20 |
| | −/− | −74 ± 5 (5) | 1000 ± 300 (4) | 4.2 ± 1.0 (4) | 4.4 ± 0.2 (14) | 202, 24–370 |

*Striolar −/− vs ES −/−: two-way ANOVA, $p = 0.006$, g 1.2; striolar −/− vs striolar +/+,+/−: two-way ANOVA, $p = 0.005$, g 1.7.
[†]−/− vs +/+: two-way ANOVA, $p < 1E−9$, g 2.3; −/− vs +/−: two-way ANOVA, $p < 1E−9$, g 3.4.
[‡]−/− vs +/+: two-way ANOVA, $p < 1E−9$, g 3.1; −/− vs +/−: two-way ANOVA, $p < 1E−9$, g 3.9.
[§][†]−/− vs +/+: two-way ANOVA, $p < 1E−9$, g 2.7; −/− vs +/−: two-way ANOVA, $p < 1E−9$, g 3.4.
[¶][‡]−/− vs +/+: two-way ANOVA, $p = 3E−7$, g 1.5; −/− vs +/−: two-way ANOVA, $p = 1.3E−4$, g 1.0; +/−vs +/+: two-way ANOVA, $p = 0.048$, g 0.6.

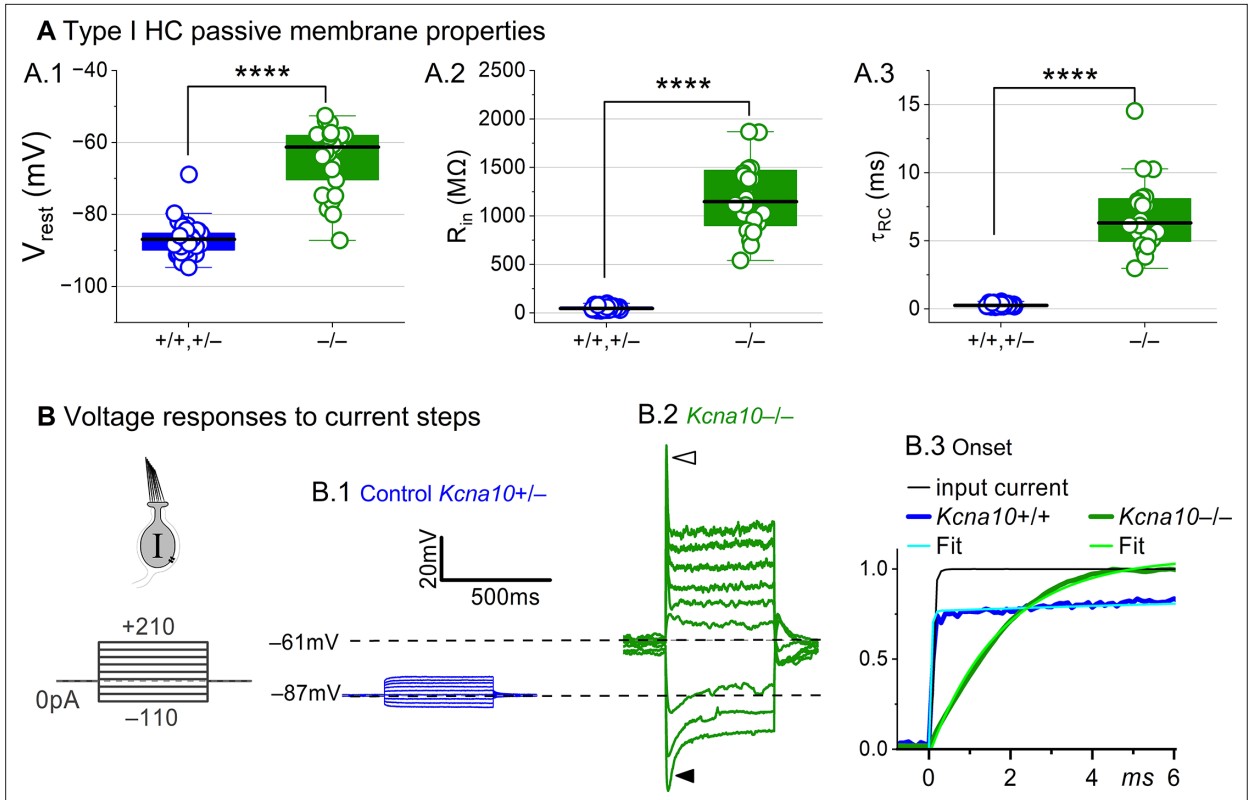

**Figure 2.** *Kcna10^{-/-}* type I hair cells (HCs) had much longer membrane charging times and higher input resistances (voltage gains) than *Kcna10^{+/+,+/-}* type I HCs. (**A**) $g_{K,L}$ strongly affects passive membrane properties: (**A.1**) $V_{rest}$, Tukey's test p<1E-9, (**A.2**) $R_{in}$, input resistance, Tukey's test p<1E-9, and (**A.3**) membrane time constant, $\tau_{RC} = (R_{input} * C_m)$, Tukey's test p<1E-9. See **Table 2** for all statistics. (**B**) Current clamp responses to the same scale from (**B.1**) *Kcna10^{+/-}* and (**B.2**) *Kcna10^{-/-}* type I cells, both P29. *Filled arrowhead (B.2),* sag indicating $I_H$ activation. *Open arrowhead*, Depolarization rapidly decays as $I_{DR}$ activates. (**B.3**) First 6 ms of voltage responses to 170 pA injection, normalized to steady-state value; *curves*, double-exponential fits (*Kcna10^{+/+}*, $\tau$ 40 µs and 2.4 ms) and single-exponential fits (*Kcna10^{-/-}*, $\tau$ 1.1 ms). *Asterisks*, ****p < 0.0001. *Line,* median; *Box,* interquartile range; *Whiskers*, outliers.

$K_V1.8$ on these properties, we switched to current clamp mode and measured resting potential ($V_{rest}$), input resistance ($R_{in}$, equivalent to voltage gain for small current steps, $\Delta V/\Delta I$), and membrane time constant ($\tau_{RC}$). In *Kcna10^{-/-}* type I HCs, $V_{rest}$ was much less negative (**Figure 2A.1**), $R_{in}$ was greater by ~20-fold (**Figure 2A.2**), and membrane charging times were commensurately longer (**Figure 2A.3**).

The differences between the voltage responses of *Kcna10^{+/+,+/-}* and *Kcna10^{-/-}* type I HCs are expected from the known impact of $g_{K,L}$ on $V_{rest}$ and $R_{in}$ (***Correia and Lang, 1990***; ***Ricci et al., 1996***; ***Rüsch and Eatock, 1996b***; ***Songer and Eatock, 2013***). The large K⁺-selective conductance at $V_{rest}$ holds $V_{rest}$ close to $E_K$ (K⁺ equilibrium potential) and minimizes gain ($\Delta V/\Delta I$), such that voltage-gated conductances are negligibly affected by the input current and the cell produces approximately linear, static responses to iterated current steps. For *Kcna10^{-/-}* type I HCs, with their less negative $V_{rest}$ and larger $R_{in}$, positive current steps evoked a fast initial depolarization (**Figure 2B.2**), activating residual delayed rectifiers and repolarizing the membrane toward $E_K$. Negative current steps evoked an initial hyperpolarization followed by a slowly repolarizing 'sag' (**Figure 2B.2**) as the HCN1 channels open (***Rüsch and Eatock, 1996b***; ***Horwitz et al., 2011***).

Overall, comparison of the *Kcna10^{+/+,+/-}* and *Kcna10^{-/-}* responses shows that with $K_V1.8$ ($g_{K,L}$), the voltage response of the type I HC is smaller but better reproduces the time course of the input current.

## $K_V1.8$ is necessary for both inactivating and non-inactivating $K_V$ currents in type II HCs

Type II HCs also express $K_V1.8$ mRNA (***McInturff et al., 2018***; ***Orvis et al., 2021***). Although their outwardly rectifying conductances ($g_A$ and $g_{DR}$) differ substantially in voltage dependence and size

from $g_{K,L}$, both conductances were strongly affected by the null mutation: $g_A$ was eliminated and the delayed rectifier was substantially smaller. Below we describe $g_A$ and $g_{DR}$ in $Kcna10^{+/+,+/-}$ type II HCs and the residual outward-rectifying current in $Kcna10^{-/-}$ type II HCs.

**$Kcna10^{+/+,+/-}$ type II HCs.** Most (81/84) $Kcna10^{+/+,+/-}$ type II HCs expressed a rapidly activating, rapidly inactivating A-type conductance ($g_A$). We define A current as the outwardly rectifying current that inactivates by over 30% within 200 ms. $g_A$ was more prominent in extrastriolar zones, as reported (**Holt et al., 1999**; **Weng and Correia, 1999**).

We compared the activation and inactivation time course and inactivation prominence for 200 ms steps from –124 to ~30 mV. Outward currents fit with **Equation 3** yielded fast inactivation time constants ($\tau_{Inact,Fast}$) of ~30 ms in LES (**Figure 3A.2**). $\tau_{Inact,Fast}$ was faster in LES than in MES or striola (**Figure 3A.3**) and fast inactivation was a larger fraction of the total inactivation in LES than striola (~0.5 vs 0.3, **Figure 3A.4**).

To show voltage dependence of activation, we generated *G–V* curves for peak currents (sum of A-current and delayed rectifier) and steady-state currents measured at 200 ms, after $g_A$ has mostly inactivated (**Figure 3D.2**). $Kcna10^{+/-}$ HCs had smaller currents than $Kcna10^{+/+}$ HCs, reflecting a smaller $g_{DR}$ (**Figure 3D**) and faster fast inactivation (**Figure 3A.3**). As discussed later, these effects may relate to effects of the $Kcna10$ gene dosage on the relative numbers of different $K_V1.8$ heteromers.

For $Kcna10^{+/+}$ and $Kcna10^{+/-}$ HCs, the voltage dependence as summarized by $V_{half}$ and slope factor (*S*) was similar. Relative to $g_{SS}$, $g_{Peak}$ had a more positive $V_{half}$ (~–21 vs ~–26) and greater *S* (~12 vs ~9, **Figure 3D**, **Table 4**). Because $g_{Peak}$ includes channels with and without fast inactivation, the shallower $g_{Peak}$–*V* curve may reflect a more heterogeneous channel population. Only $g_{Peak}$ showed zonal variation, with more positive $V_{half}$ in LES than striola (~–20 vs ~–24 mV, **Figure 3D**, **Table 4**). We later suggest that variable subunit composition may drive zonal variation in $g_{Peak}$.

**$Kcna10^{-/-}$ type II HCs.** $Kcna10^{-/-}$ type II HCs from all zones were missing $g_A$ and 30–50% of $g_{DR}$ (**Figure 3B–D**). The residual delayed rectifier (1.3 nS/pF) had a more positive $V_{half}$ than $g_{DR}$ in $Kcna10^{+/+,+/-}$ HCs (~–20 vs ~–26 mV, **Figure 3D.2**). We refer to the $K_V1.8$-dependent delayed rectifier component as $g_{DR}(K_V1.8)$ and to the residual, $K_V1.8$-independent delayed rectifier component as $g_{DR}(K_v7)$ because, as we show later, it includes $K_V7$ channels.

**Figure 3—figure supplement 2** shows the development of $K_V1.8$-dependent and -independent $K_V$ currents in type II HCs with age from P5 to over P300. In $Kcna10^{+/+,+/-}$ type II HCs, $g_A$ was present at all ages with a higher % inactivation after P18 than at P5–P10 (**Figure 3—figure supplement 2A.4**). $g_{Peak}$ did not change much above P12 except for a compression of conductance density from P13 to P370 (partial correlation coefficient = –0.4, p = 2E–5, **Figure 3—figure supplement 2A.3**).

We saw small rapidly inactivating outward currents in a minority of $Kcna10^{-/-}$ type II HCs (23%, 7/30), all >P12 and extrastriolar (**Figure 3—figure supplement 3**). These currents overlapped with $g_A$ in percent inactivation, inactivation kinetics, and activation voltage dependence but were very small. As discussed later, we suspect that these currents flow through homomers of inactivating $K_V$ subunits that in control HCs join with $K_V1.8$ subunits and confer inactivation on the heteromeric conductance.

## $K_V1.8$ affects type II passive properties and responses to current steps

In type II HCs, absence of $K_V1.8$ did not change $V_{rest}$ (**Figure 4A.1**) because $g_A$ and $g_{DR}$ both activate positive to rest, but significantly increased $R_{in}$ and $\tau_{RC}$ (**Figure 4A.2 and A.3**).

Positive current steps evoked an initial depolarizing transient in both $Kcna10^{+/+}$ and $Kcna10^{-/-}$ type II HCs, but the detailed time course differed (**Figure 4B**). Both transient and steady-state responses were larger in $Kcna10^{-/-}$, consistent with their larger $R_{in}$ values.

Absence of $K_V1.8$ increased the incidence of sharp electrical resonance in type II HCs. Electrical resonance, which manifests as ringing responses to current steps, can support receptor potential tuning (**Ashmore, 1983**; **Fettiplace, 1987**; **Hudspeth and Lewis, 1988**; **Ramanathan and Fuchs, 2002**). Larger $R_{in}$ values made $Kcna10^{-/-}$ type II HCs more prone to electrical resonance; **Figure 4C.1** shows a striking example. Median resonance quality ($Q_e$, sharpness of tuning) was greater in $Kcna10^{-/-}$ (1.33, n=26) than $Kcna10^{+/+}$ (0.66, *n* = 23) or $Kcna10^{+/-}$ (0.59, *n* = 44) type II HCs.

$K_V1.8$ affected the time course of the initial peak in response to much larger current injections (**Figure 4D, E**). Fast activation of $g_A$ in control type II HCs rapidly repolarizes the membrane and then inactivates, allowing the constant input current to progressively depolarize the cell, producing a slow

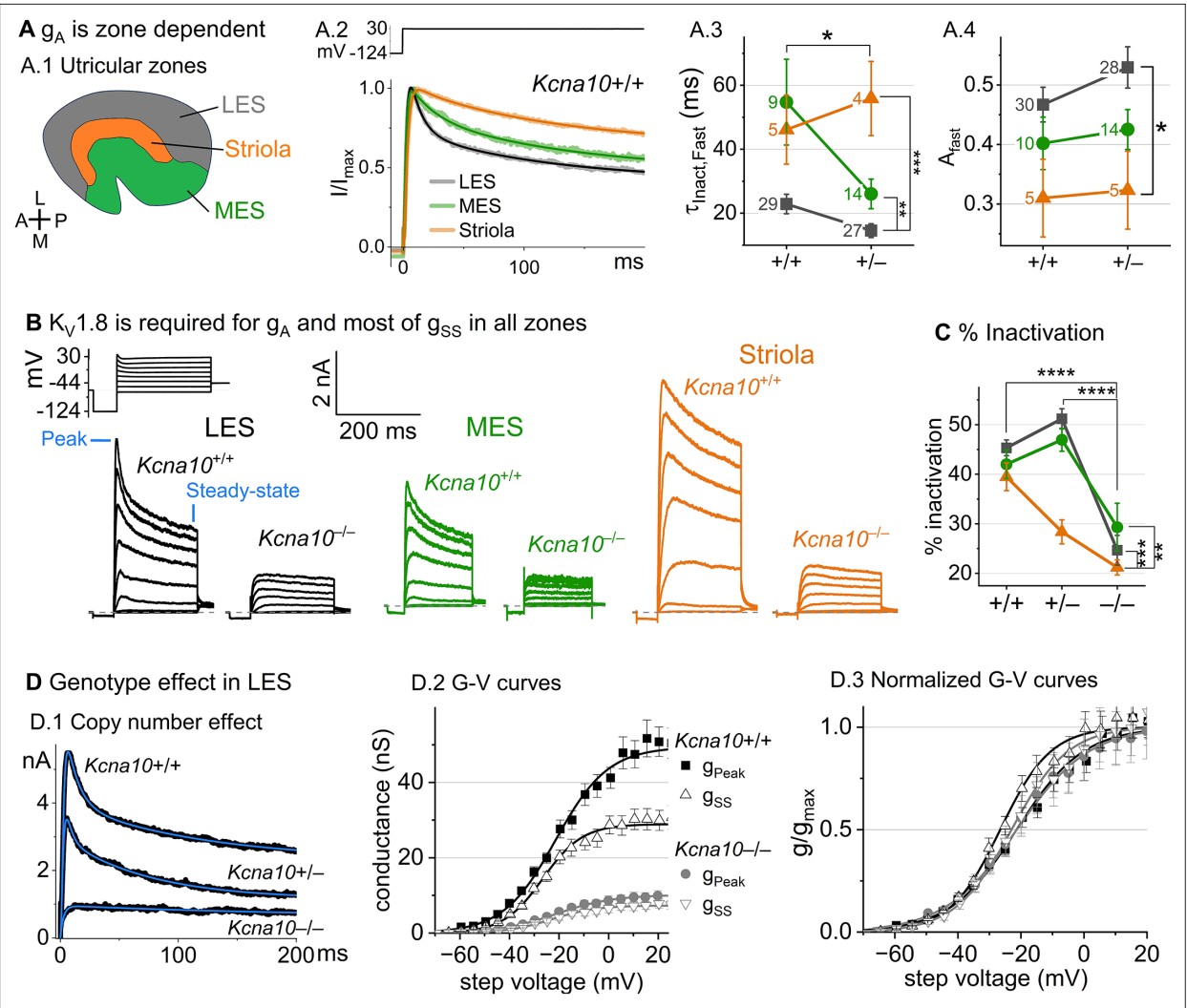

**Figure 3.** *Kcna10⁻/⁻* type II hair cells (HCs) in all zones of the sensory epithelium lacked the major rapidly inactivating conductance, $g_A$, and had less delayed rectifier conductance. Activation and inactivation varied with epithelial zone and genotype. (**A**) $g_A$ inactivation time course varied across zones. (**A.1**) Zones of the utricular epithelium: lateral extrastriola (LES), medial extrastriola (MES), and striola (S). (**A.2**) Normalized currents evoked by steps from –124 to +30 mV with overlaid fits of *Equation 3*. (**A.3**) $\tau_{Inact,Fast}$ was faster in *Kcna10⁺/⁻* (n=45) than *Kcna10⁺/⁺* (n=43) HCs (KWA, p=0.027), and faster in LES (n=56) than MES (n=23, KWA, p=0.002) or S (n=9, KWA, p=2E-4). Point label is number of cells. Brackets show post hoc pairwise comparisons between two zones (vertical brackets) and horizontal brackets compare two genotypes; see *Table 3* for statistics on kinetics. (**A.4**) Fast inactivation was a greater fraction of total inactivation in LES (n=58) than striola (n=10, Tukey's test p=0.0041). (**B**) Exemplars; ages, *left to right*, P312, P53, P287, P49, P40, P154. (**C**) % inactivation at 30 mV was much lower in *Kcna10⁻/⁻* (n=37) than *Kcna10⁺/⁻* (n=47, Tukey's HSD, p<1E-9) and *Kcna10⁺/⁺* (n=44, Tukey's HSD, p<1E-9). % inactivation was lower in striola (n=16) than LES (n=77, Tukey's HSD, p=3E-5) and MES (n=36, Tukey's HSD, p=0.0011). 2-way ANOVA detected interaction between zone and genotype, p=0.026 (*Table 3*). (**D**) Exemplar currents and *G–V* curves from LES type II HCs show a copy number effect. (**D.1**) Exemplar currents evoked by steps from –124 to +30 mV fit with *Equation 3*. (**D.2**) Averaged peak and steady-state conductance–voltage data points from LES cells (+/+, n=37; –/–, n=20) were fit with Boltzmann equations (*Equation 1*) and normalized by $g_{max}$ in (**D.3**). *Asterisks*: *p < 0.05; **p < 0.01; ***p < 0.001; and ****p < 0.0001. *Error bars*, SEM. See *Table 4* for statistics on voltage dependence.

The online version of this article includes the following figure supplement(s) for figure 3:

**Figure supplement 1.** For type II hair cells (HCs) older than P12, $K_V$ conductance activation and inactivation differed across zones and genotypes.

**Figure supplement 2.** For type II hair cells (HCs) older than P12, $K_V$ conductances were stable.

**Figure supplement 3.** A minority of extrastriolar *Kcna10⁻/⁻* type II hair cells (HCs) had a very small fast-inactivating outward rectifier current.

**Table 3.** Type II hair cell $K_V$ currents: activation and inactivation time course at +30 mV. Mean ± SEM. g is effect size, Hedge's g. KWA is Kruskal–Wallis ANOVA.

| Zone | Kcna10 | $\tau_{Act}$ at 30 mV, ms[*, †] | $\tau_{Inact,Fast}$ at 30 mV, ms [‡, §] | Fast inactivation prominence[¶] | Inactivation %[**,††] | N cells | Age (median, range) |
|---|---|---|---|---|---|---|---|
| LES | +/+ | 2.11 ± 0.09 | 23 ± 3 | 0.46 ± 0.03 | 45 ± 2 | 30 | 46, 14–312 |
| | +/– | 1.64 ± 0.09 | 15 ± 2 | 0.53 ± 0.03 | 51 ± 2 | 27 | 29, 13–280 |
| | –/– | 4.4 ± 0.5 | NA | NA | 25 ± 3 | 21 | 128, 15–355 |
| MES | +/+ | 2.8 ± 0.5 | 50 ± 10 | 0.40 ± 0.04 | 42 ± 3 | 9 | 94, 22–296 |
| | +/– | 2.2 ± 0.2 | 90 ± 60 | 0.42 ± 0.03 | 47 ± 2 | 15 | 24, 13–52 |
| | –/– | 10 ± 7 | NA | NA | 29 ± 5 | 10 | 84, 28–355 |
| Striola | +/+ | 2.7 ± 0.3 | 50 ± 10 | 0.31 ± 0.07 | 39 ± 3 | 5 | 45, 40–287 |
| | +/– | 2.9 ± 0.4 | 300 ± 200 | 0.3 ± 0.06 | 28 ± 2 | 5 | 19, 14–30 |
| | –/– | 7 ± 2 | NA | NA | 22 ± 2 | 6 | 202, 29–298 |

[*]–/– vs +/+: KWA, p = 0.0048, g 0.6; –/– vs +/–: KWA, p = 2.3E−7, g 0.6.

[†]Striola vs LES: KWA, p = 5.7E−4, g 1.0.

[‡]+/– vs +/+: KWA, p = 0.027, g 0.2.

[§]LES vs MES: KWA, p = 0.0018, g 0.3; LES vs Striola: KWA, p = 1.9E−4, g 0.8.

[¶]LES vs Striola: two-way ANOVA, p = 0.0041, g 0.7.

[**]–/– vs +/+: two-way ANOVA, p < 1E−9, g 1.7; –/– vs +/–: two-way ANOVA, p < 1E−9, g 1.8.

[††]Striola vs LES: two-way ANOVA, p = 3.4E−5, g 0.9; Striola vs MES: two-way ANOVA, p = 0.0011, g 1.0; interaction between genotype and zone: two-way ANOVA, p = 0.026.

rebound (*Figure 4D.2*). This behavior has the potential to counter mechanotranduction adaptation (*Vollrath and Eatock, 2003*).

## $K_V$1.8 immunolocalized to basolateral membranes of both type I and II HCs

If $K_V$1.8 is a pore-forming subunit in the $K_V$1.8-dependent conductances $g_{K,L}$, $g_A$, and $g_{DR}$, it should localize to HC membranes. *Figure 5* compares $K_V$1.8 immunoreactivity in *Kcna10$^{+/+}$* and *Kcna10$^{-/-}$* utricles, showing specific immunoreactivity along the basolateral membranes of both HC types in *Kcna10$^{+/+}$* utricles. To identify HC type and localize the HC membrane, we used antibodies against $K_V$7.4 (KCNQ4), an ion channel densely expressed in the calyceal 'inner-face' membrane next to the synaptic cleft (*Hurley et al., 2006*; *Lysakowski et al., 2011*), producing a cup-like stain around type I HCs (*Figure 5A*). $K_V$1.8 immunoreactivity was present in HC membrane apposing $K_V$7.4-stained calyx inner face in *Kcna10$^{+/+}$* utricles (*Figure 5A.1 and A.2*) and not in *Kcna10$^{-/-}$* utricles (*Figure 5A.3*).

In other experiments, we used antibodies against calretinin (CALB2), a cytosolic calcium-binding protein expressed by many type II HCs and also by striolar calyx-only afferents (*Desai et al., 2005*; *Lysakowski et al., 2011*, *Figure 5B*). An HC is type II if it is calretinin-positive (*Figure 5B.1*) or if it lacks a $K_V$7.4- or calretinin-positive calyceal cup (*Figure 5A.2 and B.3*, rightmost cells). HC identification was confirmed with established morphological indicators: for example, type II HCs tend to have basolateral processes (feet) (*Pujol et al., 2014*) and, in the extrastriola, more apical nuclei than type I HCs.

Previously, *Carlisle et al., 2012* reported $K_V$1.8-like immunoreactivity in many cell types in the inner ear. In contrast, *Lee et al., 2013* found that gene expression reporters indicated expression only in HCs and some supporting cells. Here, comparison of control and null tissue showed selective expression of HC membranes, and that some supporting cell staining is non-specific.

## $K_V$1.4 may also contribute to $g_A$

Results with the $K_V$1.8 knockout suggest that type II HCs have an inactivating $K_V$1 conductance that includes $K_V$1.8 subunits. $K_V$1.8, like most $K_V$1 subunits, does not show fast inactivation as a heterologously expressed homomer (*Lang et al., 2000*; *Ranjan et al., 2019*; *Dierich et al., 2020*), nor do the $K_V$1.8-dependent channels in type I HCs, as we show, and in cochlear inner HCs (*Dierich et al., 2020*).

**Table 4.** Type II hair cell $K_V$ currents: activation voltage dependence.
Mean ± SEM. g is effect size, Hedge's g. KWA is Kruskal–Wallis ANOVA.

| Zone | Kcna10 | Peak $V_{1/2}$, mV** | Peak $S$, mV[††, ‡] | A-type $g_{max}/C_m$, nS/pF[§ §] | SS $V_{half}$, mV[¶ ¶] | SS $S$, mV**** | SS $g_{max}/C_m$, nS/pF[†† ‡‡] | N cells | Age (median, range) |
|---|---|---|---|---|---|---|---|---|---|
| LES | +/+ | −19.8 ± 0.6 | 11.8 ± 0.4 | 4.0 ± 0.3 | −25.0 ± 0.5 | 8.7 ± 0.3 | 7.1 ± 0.8 | 37 | 46, 14–312 |
| | +/− | −19.8 ± 0.8 | 12.8 ± 0.4 | 3.8 ± 0.3 | −26.8 ± 0.8 | 8.7 ± 0.3 | 4.9 ±0.4 | 35 | 29, 13–280 |
| | −/− | −18 ± 1 | 11.7 ± 0.4 | 0.37 ± 0.05 | −19 ± 1 | 12.1 ± 0.5 | 1.8 ±0.2 | 20 | 128, 15–355 |
| MES | +/+ | −22 ± 1 | 11 ± 0.7 | 4.1 ± 0.7 | −26 ± 1 | 8.3 ± 0.5 | 9 ±1 | 11 | 94, 22–296 |
| | +/− | −21 ± 1 | 11.8 ± 0.4 | 3.6 ± 0.5 | −27 ± 1 | 9.0 ± 0.3 | 5.9 ±0.7 | 16 | 24, 13–52 |
| | −/− | −19 ± 1 | 10.8 ± 0.6 | 0.6 ± 0.1 | −20 ± 1 | 10.7 ± 0.7 | 2.5 ±0.3 | 15 | 84, 28–355 |
| Striola | +/+ | −24 ± 1 | 9.6 ± 0.5 | 5 ± 1 | −26.6 ± 0.9 | 8.2 ± 0.4 | 12 ±1 | 7 | 45, 40–287 |
| | +/− | −25 ± 2 | 9.4 ± 0.4 | 2.6 ± 0.6 | −28 ± 2 | 8.2 ± 0.3 | 10±2 | 6 | 19, 14–30 |
| | −/− | −21.3 ± 0.9 | 10.3 ± 0.5 | 0.7 ± 0.1 | −21.7 ± 0.8 | 10.5 ± 0.6 | 3.9±0.5 | 8 | 202, 29–298 |

*Striola vs LES: two-way ANOVA, p = 0.00116, g 0.9.
[†]Striola vs MES: two-way ANOVA, p = 0.016, g 0.8; Striola vs LES: two-way ANOVA, p = 7.5E−6, g 1.2.
[‡]−/− vs +/−: two-way ANOVA, p = 0.036, g 0.5.
[§]−/− vs +/+: Welch ANOVA, p < 1E−9, g 2.3; −/− vs +/−: Welch ANOVA, p < 1E−9, g 2.3.
[¶]−/− vs +/+: two-way ANOVA, p < 1E−9, g 1.4; −/− vs +/−: two-way ANOVA, p < 1E−9, g 1.6.
**−/− vs +/+: two-way ANOVA, p < 1E−9, g 1.4; −/− vs +/−: two-way ANOVA, p = 4.5E−7, g 1.1.
[††]−/− vs +/+: Welch ANOVA, p < 1E−9, g 1.6; −/− vs +/−: Welch ANOVA, p < 1E−9, g 1.3; +/+vs +/−: Welch ANOVA, p = 0.007, g 1.6.
[‡‡]Striola vs LES: one-way ANOVA, p = 0.001, g (0.9); Striola vs MES: one-way ANOVA, p = 0.01, g 0.8.

$K_V1$ subunits without intrinsic inactivation can produce rapidly inactivating currents by associating with $K_V\beta1$ or $K_V\beta3$ subunits. $K_V\beta1$ (*Kcnb1*) is present in type II HCs alongside $K_V\beta2$ (*Kcnb2*) (**McInturff et al., 2018**; **Jan et al., 2021**; **Orvis et al., 2021**), which does not confer rapid inactivation (**Dwenger et al., 2022**).

Another possibility is that in type II HCs, $K_V1.8$ subunits heteromultimerize with $K_V1.4$ subunits—the only $K_V1$ subunits which, when expressed as a homomer, have complete N-type (fast) inactivation (**Stühmer et al., 1989**). Multiple observations support this possibility. $K_V1.4$ has been linked to $g_A$ in pigeon type II HCs (**Correia et al., 2008**) and is the second-most abundant $K_V1$ transcript in mammalian

**Table 5.** Type II hair cell passive membrane properties in the extrastriola (ES) and striola (S).
Mean ± SEM (number of cells). g is effect size, Hedge's g. KWA is Kruskal–Wallis ANOVA. Peak height and time were measured from responses to 170 pA input from rest.

| Zone | Kcna10 | $V_{rest}$, mV | $R_{input}$, GΩ* | $\tau_{RC}$, ms[†] | Peak height, mV[‡] | Peak time, ms[§] | $C_m$, pF | Age (median, range) |
|---|---|---|---|---|---|---|---|---|
| ES | +/+ | −71 ± 2 (19) | 1.4 ± 0.2 (16) | 11 ± 3 (16) | −20 ± 2 (15) | 2.5 ± 0.2 (15) | 4.7 ± 0.2 (50) | 45, 16–312 |
| | +/− | −71 ± 2 (34) | 1.2 ± 0.1 (27) | 9 ± 1 (27) | −20 ± 1 (30) | 2.44 ± 0.08 (30) | 4.6 ± 0.1 (52) | 27, 13–280 |
| | −/− | −76 ± 2 (9) | 2.3 ± 0.3 (7) | 16 ± 3 (7) | 2 ± 6 (7) | 3.6 ± 0.3 (7) | 4.6 ± 0.2 (35) | 53, 15–154 |
| S | +/+ | −73.1 ± 1.0 (6) | 1.4 ± 0.1 (6) | 9 ± 1 (6) | −20 ± 2 (5) | 2.7 ± 0.1 (5) | 4.6 ± 0.2 (7) | 45, 40–224 |
| | +/− | −71 ± 3 (5) | 1.4 ± 0.3 (6) | 7 ± 2 (6) | −20 ± 2 (6) | 2.3 ± 0.1 (6) | 4.8 ± 0.2 (6) | 19, 19–30 |
| | −/− | −68 ± 2 (6) | 3.0 ± 0.7 (6) | 26 ± 10 (6) | 2 ± 7 (4) | 4 ± 1 (4) | 4.4 ± 0.3 (7) | 178, 29–298 |

*−/− vs +/+: KWA, p = 0.015, g 1.2; −/− vs +/−: KWA, p = 0.002, g 1.5.
[†]−/− vs +/+: KWA, p = 0.016, g 0.7; −/− vs +/−: KWA, p = 0.008, g 1.2.
[‡]−/− vs +/+: KWA, p = 0.006, g 2.1; −/− vs +/−: KWA, p = 2E−4, g 2.6.
[§]−/− vs +/+: two-way ANOVA, p < 1E−9, g 1.3; −/− vs +/−: two-way ANOVA, p < 1E−9, g 1.9.

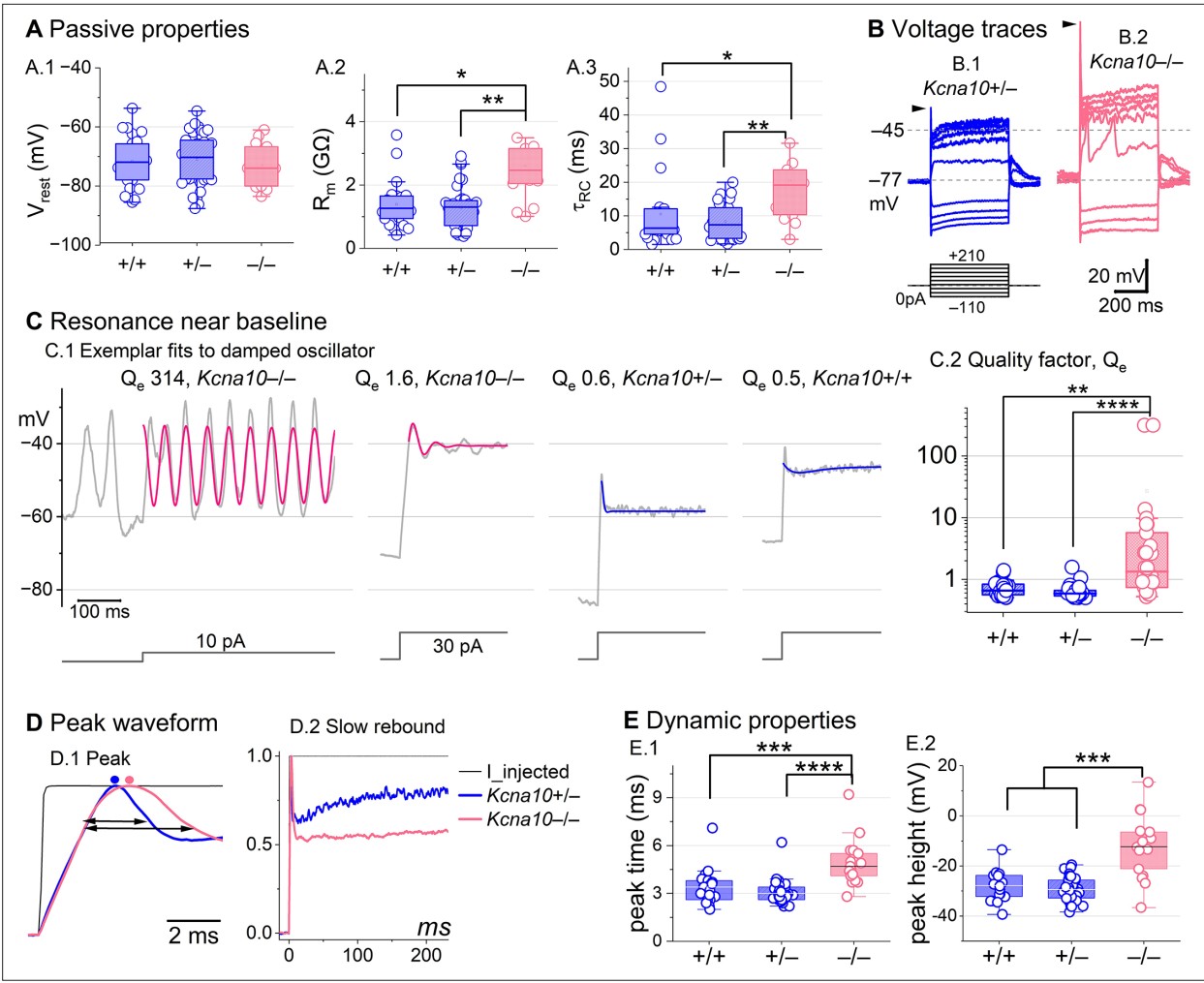

**Figure 4.** $Kcna10^{-/-}$ type II hair cells (HCs) had larger, slower voltage responses and more electrical resonance. (**A**) Passive membrane properties near resting membrane potential: (**A.1**) Resting potential. $R_{input}$ (**A.2**) and $\tau_{RC}$ (**A.3**) were obtained from single-exponential fits to voltage responses <15 mV. $R_{input}$ and $\tau_{RC}$ were higher in $Kcna10^{-/-}$ (n=13) than $Kcna10^{+/+}$ (n=22, KWA p=0.015; p=0.016) and $Kcna10^{+/-}$ (n=33, KWA p=0.002; p=0.008; see **Table 5**). (**B**) Exemplar voltage responses to iterated current steps (*bottom*) illustrate key changes in gain and resonance with $K_V1.8$ knockout. (**B.1**) $Kcna10^{+/-}$ type II HC (P24, LES) and (**B.2**) $Kcna10^{-/-}$ type II HC (P53, LES). Arrowheads, depolarizing transients. (**C**) Range of resonance illustrated for $Kcna10^{-/-}$ type II HCs (*left, pink curves fit to* **Equation 5**) and controls (*right, blue fits*). (**C.1**) *Resonant frequencies, left to right:* 19.6, 18.4, 34.4, and 0.3 Hz. Leftmost cell resonated spontaneously (before step). (**C.2**) Tuning quality ($Q_e$; **Equation 6**) was higher for $Kcna10^{-/-}$ (n=26) type II HCs (KWA: p = 0.0064 vs $Kcna10^{+/+}$, n=23; p = 7E-8 vs $Kcna10^{+/-}$, n=45). (**D**) $Kcna10^{-/-}$ type II HCs had higher, slower peaks and much slower rebound potentials in response to large (170 pA) current steps. (**D.1**) Normalized to show initial depolarizing transient (*filled circles*, times of peaks; *horizontal arrows*, peak width at half-maximum). (**D.2**) Longer time scale to highlight how null mutation reduced post-transient rebound. (**E**) In $Kcna10^{-/-}$ HCs (n=19), depolarizing transients evoked by a +90 pA step were slower to peak (**E.1**) than in $Kcna10^{+/+}$ (n=19, 2-way ANOVA Tukey's p<1E-9) and $Kcna10^{+/-}$ (n=34, 2-way ANOVA Tukey's p<1E-9) and (**E.2**) larger than in $Kcna10^{+/+}$ (n=19, KWA p=0.006) and $Kcna10^{+/-}$ (n=34, KWA p=2E-4). *Asterisks*: *p < 0.05; **p < 0.01; ***p < 0.001; and ****p < 0.0001. *Line*, median; *Box*, interquartile range; *Whiskers*, outliers.

vestibular HCs, after $K_V1.8$ (*Scheffer et al., 2015*). $K_V1.4$ is expressed in type II HCs but not type I HCs (*McInturff et al., 2018*; *Orvis et al., 2021*), and is not found in striolar HCs (*Jan et al., 2021*; *Orvis et al., 2021*), where even in type II HCs, inactivation is slower and less extensive (*Figure 3A*).

Functional heteromers form between $K_V1.4$ and other $K_V1.x$ and/or $K_Vβ1$ (*Imbrici et al., 2006*; *Correia et al., 2008*; *Al Sabi et al., 2011*). Although $K_V1.4$ and $K_V1.8$ heteromers have not been studied directly, $g_A$'s inactivation time course ($\tau_{Fast,Inact}$ of ~30 ms +30 mV, *Figure 3A*) and voltage dependence ($V_{half}$ –41 mV, *Figure 6B*) are consistent with these other $K_V1.4$-containing heteromers.

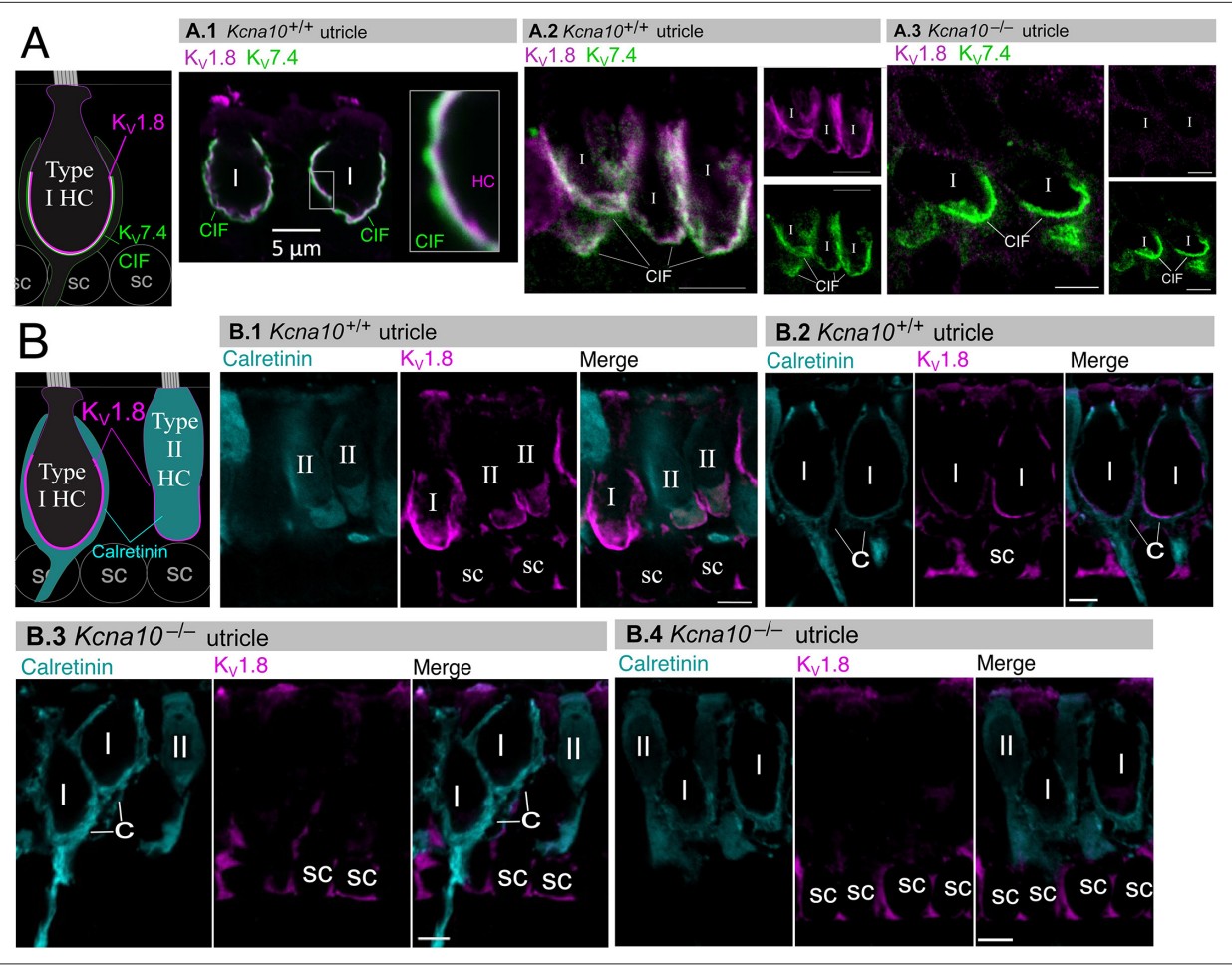

**Figure 5.** Type I and II hair cell (HC) basolateral membranes show specific immunoreactivity to Kv1.8 antibody (magenta). Antibodies for $K_V7.4$ (A, green) and calretinin (B, cyan) were used as counterstains for calyx membrane (Kv7.4), type II HC cytoplasm (calretinin) and cytoplasm of striolar calyx-only afferents (calretinin). (**A**) *Left*, Cartoon showing $K_V7.4$ on the calyx inner face membrane (CIF) and $K_V1.8$ on the type I HC membrane. SC, supporting cell nuclei. *A.1–3*, Adult mouse utricle sections. $K_V7.4$ antibody labeled calyces on two $K_V1.8$-positive type I HCs (*A.1*), four $K_V1.8$-positive type I HCs (*A.2*), and two $K_V1.8$-negative type I HCs from a *Kcna10⁻/⁻* mouse (*A.3*). (**B**) *Left*, Cartoon showing cytoplasmic calretinin stain in calyx-only striolar afferents and most type II HCs, and $K_V1.8$ on membranes of both HC types. In wildtype utricles, $K_V1.8$ immunolocalized to basolateral membranes of type I and II HCs (extrastriola, *B.1*). $K_V1.8$ immunolocalized to type I HCs (striola, *B.2*). Staining of supporting cell (SC) membranes by Kv1.8 antibody was non-specific, as it was present in *Kcna10⁻/⁻* tissue (striola, *B.3* and *B.4*). All scale bars 5 µm.

## $K_V7$ channels contribute a small delayed rectifier in type I and II HCs

In *Kcna10⁻/⁻* HCs, absence of $I_{K,L}$ and $I_A$ revealed smaller delayed rectifier $K^+$ currents that, unlike $I_{K,L}$, activated positive to resting potential and, unlike $I_A$, lacked fast inactivation. Candidate channels include members of the $K_V7$ (KCNQ, M-current) family, which have been identified previously in rodent vestibular HCs (*Kharkovets et al., 2000*; *Rennie et al., 2001*; *Hurley et al., 2006*; *Scheffer et al., 2015*).

We tested for $K_V7$ contributions in *Kcna10⁻/⁻* type I HCs, *Kcna10⁻/⁻* type II HCs, and *Kcna10⁺/⁺,⁺/⁻* type II HCs of multiple ages by applying XE991 at 10 µM (*Figure 7A*), a dose selective for $K_V7$ channels (*Brown et al., 2002*) and close to the $IC_{50}$ (*Alexander et al., 2019*). In *Kcna10⁻/⁻* HCs of both types, 10 µM XE991 blocked about half of the residual $K_V$ conductance (*Figure 7B.1*), consistent with $K_V7$ channels conducting most or all of the non-$K_V1.8$ delayed rectifier current. In all tested HCs (P8–355, median P224), the XE991-sensitive conductance did not inactivate substantially within 200 ms at any voltage, consistent with $K_V7.2$, 7.3, 7.4, and 7.5 currents (*Wang, 1998*; *Kubisch et al., 1999*; *Schroeder et al., 2000*; *Jensen et al., 2007*; *Xu et al., 2007*). We refer to this component as $g_{DR}(K_V7)$. The voltage dependence and $g_{max}$ density ($g_{max}/C_m$) of $g_{DR}(K_V7)$ were comparable across HC types

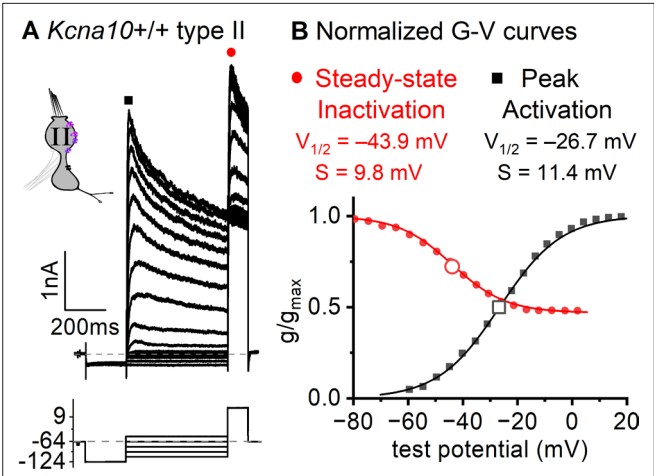

**Figure 6.** Inactivation curve of $g_A$ in extrastriolar type II hair cells (HCs). (**A**) Modified voltage protocol measured accumulated steady-state inactivation at the tail potential. 100 μM ZD7288 in bath prevented contamination by HCN current. (**B**) Voltage dependence of $g_A$'s steady-state inactivation ($h_\infty$ curve) and peak activation are consistent with $K_V$1.4 heteromers. *Curves*, Boltzmann fits (*Equation 1*). *Average fit parameters from Kcna10$^{+/+,+/-}$ type II HCs,* P40–P210, median P94. Inactivation: $V_{half}$, $-42 \pm 2$ mV ($n = 11$); $S$, $11 \pm 1$ mV. Activation: $V_{half}$, $-23 \pm 1$ mV ($n = 11$); $S$, $11.2 \pm 0.4$ mV.

and genotypes (*Figure 7B.2–4*). Although $K_V$7.4 was not detectable in HCs during immunostaining (*Figure 5*), $K_V$7.4 has been shown in type I HCs with immunogold labeling (*Kharkovets et al., 2000*; *Hurley et al., 2006*).

These results are consistent with similar $K_V$7 channels contributing a relatively small delayed rectifier in both HC types. In addition, the similarity of XE991-sensitive currents of *Kcna10$^{+/+}$* and *Kcna10$^{-/-}$* type II HCs indicates that knocking out $K_V$1.8 did not cause general effects on ion channel expression. We did not test XE991 on *Kcna10$^{+/+,+/-}$* type I HCs because $g_{K,L}$ runs down in ruptured patch recordings (*Rüsch and Eatock, 1996a*; *Chen and Eatock, 2000*; *Hurley et al., 2006*), which could contaminate the XE991-sensitive conductance obtained by subtraction.

In one striolar *Kcna10$^{-/-}$* type I HC, XE991 also blocked a small conductance that activated negative to rest (*Figure 7—figure supplement 1A, B*). This conductance ($V_{half} \sim = -100$ mV, *Figure 7—figure supplement 1C*) was detected only in *Kcna10$^{-/-}$* type I HCs from the striola (5/23 vs 0/45 extrastriolar). The $V_{half}$ and $\tau_{deactivation}$ were similar to values reported for $K_V$7.4 channels in cochlear HCs (*Wong et al., 2004*; *Dierich et al., 2020*). This very negatively activating $K_V$7 conductance coexisted with the larger more positively activating $K_V$7 conductance (*Figure 7—figure supplement 1C*) and was too small (<0.5 nS/pF) to contribute significantly to $g_{K,L}$ (~10–100 nS/pF, *Figure 1D*).

## Other channels

While our data are consistent with $K_V$1.8- and $K_V$7-containing channels carrying most of the outward-rectifying current in mouse utricular HCs, there is evidence in other preparations for additional channels, including $K_V$11 (KCNH, Erg) channels in rat utricular type I HCs (*Hurley et al., 2006*) and BK (KCNMA1) channels in rat utricle and rat and turtle semicircular canal HCs (*Schweizer et al., 2009*; *Contini et al., 2020*).

BK is expressed in mouse utricular HCs (*McInturff et al., 2018*; *Jan et al., 2021*; *Orvis et al., 2021*). However, Ca$^{2+}$-dependent currents have not been observed in mouse utricular HCs, and we found little to no effect of the BK-channel blocker iberiotoxin at a dose (100 nM) well beyond the IC$_{50}$: percent blocked at −30 mV was $2 \pm 6\%$ (3 *Kcna10$^{-/-}$* type I HCs); $1 \pm 5\%$ (5 *Kcna10$^{+/+,+/-}$* type II HCs); 7% and 14% (2 *Kcna10$^{-/-}$* type II HCs). We also did not see N-shaped *I–V* curves typical of many Ca$^{2+}$-dependent K$^+$ currents. In our ruptured-patch recordings, Ca$^{2+}$-dependent BK currents and erg channels may have been eliminated by wash-out of the HCs' small Ca$_V$ currents (*Bao et al., 2003*) or cytoplasmic second messengers (*Hurley et al., 2006*).

To check whether the constitutive $K_V$1.8 knockout has strong non-specific effects on channel trafficking, we examined the summed HCN and fast inward rectifier currents ($I_H$ and $I_{Kir}$) at −124 mV, and

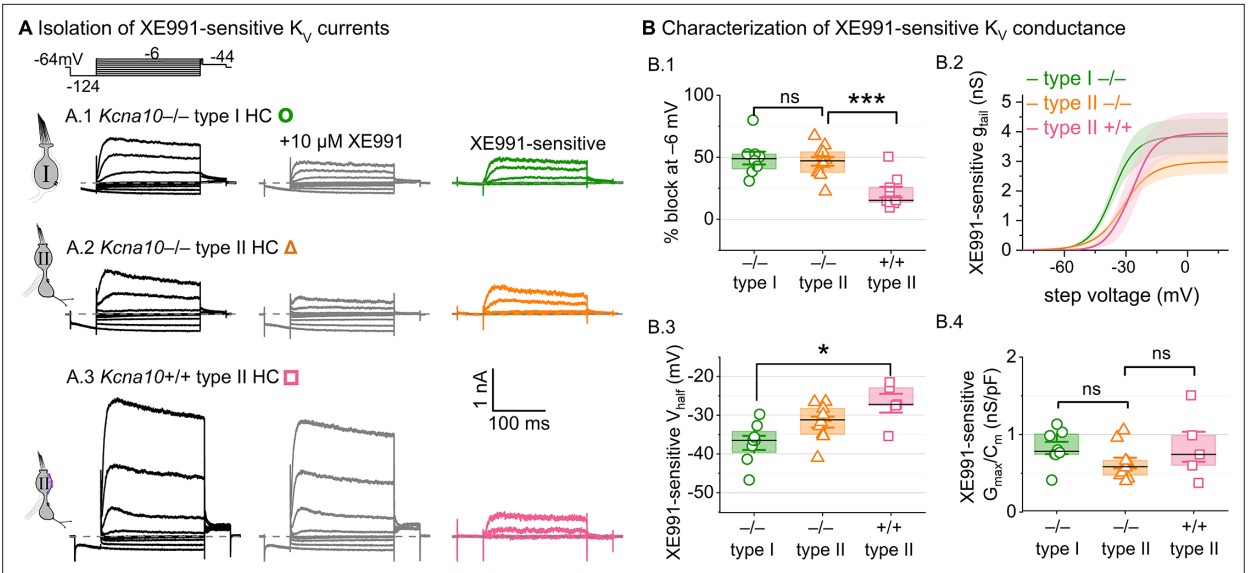

**Figure 7.** A $K_V7$-selective blocker, XE991, reduced residual delayed rectifier currents in *Kcna10⁻/⁻* type I and II hair cells (HCs). (**A**) XE991 (10 μM) partly blocked similar delayed rectifier currents in type I and II *Kcna10⁻/⁻* HCs and a type II *Kcna10⁺/⁺* HC. (**B**) Properties of XE991-sensitive conductance, $_{DR}(K_V7)$. (**B.1**) % Block of steady-state current. (**B.2**) Mean tail *G–V* curves for *Kcna10⁻/⁻* type I HCs ($n$ = 8), *Kcna10⁻/⁻* type II HCs (9), and *Kcna10⁺/⁺* type II HCs (5); shading is ± SEM. (**B.3**) $V_{half}$ was less negative in *Kcna10⁺/⁺* type II than *Kcna10⁻/⁻* type I HC ($p$ = 0.01, KWA). (**B.4**) Conductance density was similar in all groups (ANOVA), non-significant at 0.4 power (*left*), 0.2 power (*right*). *Asterisks*: *$p$ < 0.05 and ***$p$ < 0.001. *Line,* median; *Box,* interquartile range; *Whiskers,* outliers.

The online version of this article includes the following figure supplement(s) for figure 7:

**Figure supplement 1.** A minority of striolar *Kcna10⁻/⁻* type I hair cells (HCs) had a small low-voltage-activated outward rectifier current in addition to a more positively activating outward rectifier.

**Figure supplement 2.** No difference was detected in H (HCN) and Kir (fast inward rectifier) currents between *Kcna10⁺/⁺* and *Kcna10⁻/⁻* hair cells (HCs), consistent with a specific involvement of $K_V1.8$ in *Kcna10* expression.

found them similar across genotypes (*Figure 7—figure supplement 2*). The $g_{K,L}$ knockout allowed identification of zonal differences in $I_H$ and $I_{Kir}$ in type I HCs, previously examined in type II HCs (*Masetto and Correia, 1997*; *Levin and Holt, 2012*). In type I HCs from both control and null utricles, $I_H$ and $I_{Kir}$ were less prevalent in striola than extrastriola, and, when present, the combined inward current was smaller (*Figure 7—figure supplement 2*).

## Discussion

We have shown that constitutive knockout of $K_V1.8$ eliminated $g_{K,L}$ in type I HCs, and $g_A$ and much of $g_{DR}$ in type II HCs. $K_V1.8$ immunolocalized specifically to the basolateral membranes of type I and II HCs. We conclude that $K_V1.8$ is a pore-forming subunit of $g_{K,L}$, $g_A$, and part of $g_{DR}$ [$g_{DR}(K_V1.8)$]. We suggest that fast inactivation of $g_A$ may arise from heteromultimerization of non-inactivating $K_V1.8$ subunits and inactivating $K_V1.4$ subunits. Finally, we showed that a substantial component of the residual delayed rectifier current in both type I and II HCs comprises $K_V7$ channels.

$K_V1.8$ is expressed in HCs from mammalian cochlea (*Dierich et al., 2020*), avian utricle (*Scheibinger et al., 2022*), and zebrafish (*Erickson and Nicolson, 2015*). Our work suggests that in anamniotes, which lack type I cells and $g_{K,L}$, $K_V1.8$ contributes to $g_A$ and $g_{DR}$, which are widespread in vertebrate HCs (reviewed in *Meredith and Rennie, 2016*). $K_V1.8$ expression has not been detected in rodent brain but is reported in the pacemaker nucleus of weakly electric fish (*Smith et al., 2018*).

### $K_V1.8$ subunits may form homomultimers to produce $g_{K,L}$ in type I HCs

Recent single-cell expression studies on mouse utricles (*McInturff et al., 2018*; *Jan et al., 2021*; *Orvis et al., 2021*) have detected just one $K_V1$ subunit, $K_V1.8$, in mouse type I HCs. Given that $K_V1.8$ can only form multimers with $K_V1$ family members, and given that $g_{K,L}$ channels are present at very

high density (~150 per μm² in rat type I, **Chen and Eatock, 2000**), it stands to reason that most or all of the channels are $K_V1.8$ homomers. Other evidence is consistent with this proposal. $g_{K,L}$ (**Rüsch and Eatock, 1996a**) and heterologously expressed $K_V1.8$ homomers in oocytes (**Lang et al., 2000**) are non-inactivating and blocked by millimolar $Ba^{2+}$ and 4-aminopyridine and >10 mM tetraethyl ammonium. Unlike channels with $K_V1.1$, $K_V1.2$, and $K_V1.6$ subunits, $g_{K,L}$ is not sensitive to 10 nM α-dendrotoxin (**Rüsch and Eatock, 1996a**). $g_{K,L}$ and heterologously expressed $K_V1.8$ channels have similar single-channel conductances (~20 pS for $g_{K,L}$ at positive potentials, **Chen and Eatock, 2000**; 11 pS in oocytes, **Lang et al., 2000**). $g_{K,L}$ is inhibited—or positively voltage-shifted—by cGMP (**Behrend et al., 1997**; **Chen and Eatock, 2000**), presumably via the C-terminal cyclic nucleotide-binding domain of $K_V1.8$.

A major novel property of $g_{K,L}$ is that it activates 30–60 mV negative to type II $K_V1.8$ conductances and most other low-voltage-activated $K_V$ channels (**Ranjan et al., 2019**). The very negative activation range is a striking difference between $g_{K,L}$ and known homomeric $K_V1.8$ channels. Heterologously expressed homomeric $K_V1.8$ channels have an activation $V_{half}$ of –10 to 0 mV (*X. laevis* oocytes, **Lang et al., 2000**; Chinese hamster ovary cells, **Dierich et al., 2020**). In cochlear inner HCs, currents attributed to $K_V1.8$ (by subtraction of other candidates) have a near-zero activation $V_{half}$ (–4 mV, **Dierich et al., 2020**).

Possible factors in the unusually negative voltage dependence of $g_{K,L}$ include:

(1) *Elevation of extracellular $K^+$* by the enveloping calyceal terminal, unique to type I HCs (**Lim et al., 2011**; **Contini et al., 2012**; **Spaiardi et al., 2020**; **Govindaraju et al., 2023**). High $K^+$ increases conductance though $g_{K,L}$ channels (**Contini et al., 2020**), perhaps through $K^+$-mediated relief of C-type inactivation (**López-Barneo et al., 1993**; **Baukrowitz and Yellen, 1995**). We note, however, that $g_{K,L}$ is open at rest even in neonatal mouse utricles cultured without innervation (**Rüsch et al., 1998**) and persists in dissociated type I HCs (**Chen and Eatock, 2000**; **Hurley et al., 2006**).

(2) *The high density of $g_{K,L}$* (~50 nS/pF in striolar $Kcna10^{+/+}$ HCs) implies close packing of channels, possibly represented by particles (12–14 nm) seen in freeze-fracture electron microscopy of the type I HC membrane (**Gulley and Bagger-Sjöbäck, 1979**; **Sousa et al., 2009**). Such close channel packing might hyperpolarize in situ voltage dependence of $g_{K,L}$, as proposed for $K_V7.4$ channels in outer HCs (**Perez-Flores et al., 2020**). Type I HC-specific partners that may facilitate this close packing include ADAM11 (**McInturff et al., 2018**), which clusters presynaptic $K_V1.1$ and $K_V1.2$ to enable ephaptic coupling at a cerebellar synapse (**Kole et al., 2015**).

(3) *Modulation by accessory subunits.* Type I HCs express $K_Vβ1$ (**McInturff et al., 2018**; **Orvis et al., 2021**), an accessory subunit that can confer fast inactivation and hyperpolarize activation $V_{half}$ by ~10 mV. $K_Vβ1$ might interact with $K_V1.8$ to shift voltage dependence negatively. Arguments against this possibility include that $g_{K,L}$ lacks fast inactivation (**Rüsch and Eatock, 1996a**; **Hurley et al., 2006**; **Spaiardi et al., 2017**) and that cochlear inner HCs co-express $K_V1.8$ and $K_Vβ1$ (**Liu et al., 2018**) but their $K_V1.8$ conductance has a near-0 $V_{half}$ (**Dierich et al., 2020**).

## $K_V1.8$ subunits may combine with different subunits to produce $g_A$ and $K_V1.8$-dependent $g_{DR}$ in type II HCs

The $K_V1.8$-dependent conductances of type II HCs vary in their fast and slow inactivation. In not showing fast inactivation (**Lang et al., 2000**; **Ranjan et al., 2019**; **Dierich et al., 2020**), heterologously expressed $K_V1.8$ subunits resemble most other $K_V1$ family subunits, with the exception of $K_V1.4$ (for comprehensive review, see **Ranjan et al., 2019**). $K_V1.4$ is a good candidate to provide fast inactivation based on immunolocalization and voltage dependence (**Figures 4 and 6**). We suggest that $g_A$ and $g_{DR}(K_V1.8)$ are $K_V1.8$-containing channels that may include a variable number of $K_V1.4$ subunits and $K_Vβ2$ and $K_Vβ1$ accessory subunits.

$K_V1.4$–$K_V1.8$ heteromeric assembly could account for several related observations. The faster $τ_{Inact,Fast}$ in $Kcna10^{+/-}$ relative to $Kcna10^{+/+}$ type II HCs (**Figure 3A.3**, **Figure 3—figure supplement 1A.1**) could reflect an increased ratio of $K_V1.4$–$K_V1.8$ subunits and therefore more N-terminal inactivation domains per heteromeric channel. Zonal variation in the extent and speed of N-type inactivation (**Figure 3A**) might arise from differential expression of $K_V1.4$. The small fast-inactivating conductance in ~20% of extrastriolar $Kcna10^{-/-}$ type II HCs (**Figure 3—figure supplement 3**) might flow through $K_V1.4$ homomers.

Fast inactivation may also receive contribution from $K_V\beta$ subunits. $K_V\beta1$ is expressed in type II HCs (*McInturff et al., 2018*; *Jan et al., 2021*; *Orvis et al., 2021*), and, together with $K_V1.4$, has been linked to $g_A$ in pigeon vestibular HCs (*Correia et al., 2008*). $K_V\beta2$, also expressed in type II HCs (*McInturff et al., 2018*; *Orvis et al., 2021*), accelerates but does not confer fast inactivation.

We speculate that $g_A$ and $g_{DR}(K_V1.8)$ have different subunit composition: $g_A$ may include heteromers of $K_V1.8$ with other subunits that confer rapid inactivation, while $g_{DR}(K_V1.8)$ may comprise homomeric $K_V1.8$ channels, given that they do not have N-type inactivation.

### $K_V1.8$ relevance for vestibular function

In both type I and II utricular HCs, $K_V1.8$-dependent channels strongly shape receptor potentials in ways that promote temporal fidelity rather than electrical tuning (*Lewis, 1988*), consistent with the utricle's role in driving reflexes that compensate for head motions as they occur. This effect is especially pronounced for type I HCs, where the current-step evoked voltage response reproduces the input with great speed and linearity (*Figure 2*).

$g_{K,L}$dominates passive membrane properties in mature $Kcna10^{+/+,+/-}$ type I HCs such that $Kcna10^{-/-}$ type I HCs are expected to have receptor potentials with higher amplitudes but lower low-pass corner frequencies, closer to those of type II HCs and immature HCs of all types (*Correia et al., 1996*; *Rüsch and Eatock, 1996a*; *Songer and Eatock, 2013*). In $Kcna10^{-/-}$ epithelia, we expect the lack of a large basolateral conductance open at rest to reduce the speed and gain of non-quantal transmission, which depends on $K^+$ ion efflux from the type I HC to change electrical and $K^+$ potentials in the synaptic cleft (*Govindaraju et al., 2023*). In HCs, $K^+$ enters the mechanosensitive channels of the hair bundle from the $K^+$-rich apical endolymph and exits through basolateral potassium conductances into the more conventional low-$K^+$ perilymph. For the type I-calyx synapse, having in the HC a large, non-inactivating $K^+$ conductance open across the physiological range of potentials avoids channel gating time and allows for instantaneous changes in current into the cleft and fast afferent signaling (*Pastras et al., 2023*).

In contrast, mature type II HCs face smaller synaptic contacts and have $K_V1.8$-dependent currents that are not substantially activated at resting potential. They do affect the time course and gain of type II HC responses to input currents, speeding up depolarizing transients, producing a repolarizing rebound during the step, and reducing resonance.

Type I and II vestibular HCs are closely related, such that adult type II HCs acquire type I-like properties upon deletion of the transcription factor *Sox2* (*Stone et al., 2021*). In normal development of the two cell types, the *Kcna10* gene generates biophysically distinct and functionally different ion channels, presenting a natural experiment in functional differentiation of sensory receptor cells.

## Materials and methods

### Key resources table

| Reagent type (species) or resource | Designation | Source or reference | Identifiers | Additional information |
|---|---|---|---|---|
| Antibody | Anti-Kv1.8 (Rabbit polyclonal) | Alomone | Cat# APC-157, lot# 0102, RRID:AB_2341039 | 1:200 or 1:400 |
| Antibody | Anti-calretinin (goat polyclonal) | Millipore | Cat# AB1550, lot# 9669, RRID:AB_90764 | 1:600 |
| Antibody | Anti-Kv7.4 (mouse IgG1 monoclonal) | NeuroMab | Cat# 2HK-65, RRID:AB_2131828 | 1:200 |
| Peptide, recombinant protein | Iberiotoxin | Alomone | STI-400 | 100 nM (water) |
| Chemical compound, drug | XE991 | Sigma | X2254 | 100 µM (water) |
| Chemical compound, drug | ZD7288 | Tocris | APN18035-2 | 100 µM (water) |
| Peptide, recombinant protein | Bovine serum albumin | Fisher | BP671 | 1 mg/ml (water) |

### Preparation

All procedures for handling animals followed the NIH Guide for the Care and Use of Laboratory Animals and were approved by the Institutional Animal Care and Use Committees of the University

of Chicago (Animal Care and Use Procedure #72360) and the Office of Animal Care and Institutional Biosafety at the University of Illinois Chicago (Protocol for Animal Use #17106). Most mice belonged to a transgenic line with a knockout allele of *Kcna10* (referred to here as *Kcna10^{-/-}*). Our breeding colony was established with a generous gift of such animals from Sherry M. Jones and Thomas Friedman. These animals are described in their paper (**Lee et al., 2013**). Briefly, the Texas A&M Institute for Genomic Medicine generated the line on a C57BL/6;129SvEv mixed background by replacing Exon 3 of the *Kcna10* gene with an IRES-bGeo/Purocassette. Mice in our colony were raised on a 12:12 hr light–dark cycle with access to food and water ad libitum.

Semi-intact utricles were prepared from ~150 male and ~120 female mice, postnatal days (P) 5–375, for same-day recording. HC $K_V$ channel data were pooled across sexes as most results did not appear to differ by sex; an exception was that $g_{K,L}$ had a more negative $V_{half}$ in males (**Supplementary file 1a**), an effect not clearly related to age, copy number, or other properties of the activation curve.

Preparation, stimulation, and recording methods followed our previously described methods for the mouse utricle (**Vollrath and Eatock, 2003**). Mice were anesthetized through isoflurane inhalation. After decapitation, each hemisphere was bathed in ice-cold, oxygenated Liebowitz-15 (L15) media. The temporal bone was removed, the labyrinth was cut to isolate the utricle, and the nerve was cut close to the utricle. The utricle was treated with proteinase XXIV (100 µg/ml, ~10 min, 22°C) to facilitate removal of the otoconia and attached gel layer and mounted beneath two glass rods affixed at one end to a coverslip.

## Electrophysiology

We used the HEKA Multiclamp EPC10 with Patchmaster acquisition software, filtered by the integrated HEKA filters: a 6-pole Bessel filter at 10 kHz and a second 4-pole Bessel filter at 5 kHz, and sampled at 10–100 kHz. Recording electrodes were pulled (PC-100, Narishige) from soda lime glass (King's Precision Glass R-6) wrapped in paraffin to reduce pipette capacitance. Internal solution contained (in mM) 135 KCl, 0.5 MgCl$_2$, 3 MgATP, 5 4-(2-hydroxyethyl)piperazine-1-ethane-sulfonic acid (HEPES), 5 ethylene glycol tetraacetic acid (EGTA), 0.1 CaCl$_2$, 0.1 Na-cAMP, 0.1 LiGTP, 5 Na$_2$CreatinePO$_4$ adjusted to pH 7.25 and ~280 mmol/kg by adding ~30 mM KOH. External solution was Liebowitz-15 media supplemented with 10 mM HEPES (pH 7.40, 310 ± 10 mmol/kg). Recording temperature was 22–25°C. Pipette capacitance and membrane capacitance transients were subtracted during recordings with Patchmaster software. Series resistance (8–12 MΩ) was measured after rupture and compensated 60–80% with the amplifier, to final values of ~2 MΩ. Potentials are corrected for remaining (uncompensated) series resistance and liquid junction potential of ~+4 mV, calculated with LJPCalc software (**Marino et al., 2014**).

*Kcna10^{-/-}* HCs appeared healthy in that cells had resting potentials negative to –50 mV, cells lasted a long time (20–30 min) in ruptured patch recordings, membranes were not fragile, and extensive blebbing was not seen. Type I HCs with $g_{K,L}$ were transiently hyperpolarized to ~–90 mV to close $g_{K,L}$ enough to increase $R_{input}$ above 100 MΩ, as needed to estimate series resistance and cell capacitance. The average resting potential, $V_{rest}$, was –87 mV ±1 (41), similar to the calculated $E_K$ of –86.1 mV, which is not surprising given the large K$^+$ conductance of these cells. $V_{rest}$ is likely more positive in vivo, where lower endolymphatic Ca$^{2+}$ increases standing inward current through MET channels.

Voltage protocols to characterize $K_V$ currents differed slightly for type I and II HCs. In standard protocols, the cell is held at a voltage near resting potential (–74 mV in type I and –64 mV in type II), then jumped to –124 mV for 200 ms in type I HCs in order to fully deactivate $g_{K,L}$ and 50 ms in type II HCs in order to remove baseline inactivation of $g_A$. The subsequent iterated step depolarizations lasted 500 ms in type I HCs because $g_{K,L}$ activates slowly (**Wong et al., 2004**) and 200 ms in type II HCs, where $K_V$ conductances activate faster. The 50 ms tail voltage was near the reversal potential of HCN channels (–44 mV in mouse utricular HCs, **Rüsch et al., 1998**) to avoid HCN current contamination.

*G–V* (activation) parameters for control type I cells may be expected to vary across experiments on semi-intact (as here), organotypically cultured and denervated (**Rüsch et al., 1998**), or dissociated-cell preparations, reflecting variation in retention of the calyx (Discussion) and voltage step durations (**Wong et al., 2004**) which elevate K$^+$ concentration around the HC. Nevertheless, the values we obtained for type I and II HCs resemble values recorded elsewhere, including experiments in which extra care was taken to avoid extracellular K$^+$ accumulation (**Spaiardi et al., 2017**; **Spaiardi et al., 2020**). The effects of K$^+$ accumulation on $g_{K,L}$'s steady-state activation curves are

small because the operating range is centered on E and can be characterized with relatively small currents (*Figure 1A*).

## Pharmacology

Drug-containing solutions were locally with BASI Bee Hive syringes at a final flow rate of 20 µl/min and a dead time of ~30 s. Global bath perfusion was paused during drug perfusion and recording, and only one cell was used per utricle. Aliquots of test agents in solution were prepared, stored at –20°C, and thawed and added to external solution on the recording day (see Key Resources Table).

## Analysis

Data analysis was performed with software from OriginLab (Northampton, MA) and custom MATLAB scripts using MATLAB fitting algorithms.

## Fitting voltage dependence and time course of conductances

*G–V curves.* Current was converted to conductance (*G*) by dividing by driving force (*V* – $E_K$; $E_K$ calculated from solutions). For type I HCs, tail *G–V* curves were generated from current 1 ms after the end of the iterated voltage test step. For type II HCs, peak *G–V* curves were generated from peak current during the step and steady-state *G–V* curves were generated from current 1 ms before the end of a 200-ms step. We fit *G–V* curves to the first-order Boltzmann equation (*Equation 1*) using a custom MATLAB function (fitzmann.m, *Source code 2*).

$$G\left(V\right) = G_{min} + \frac{G_{max}}{1 + exp(\frac{V_{half} - V}{S})} \tag{1}$$

$V_{half}$ is the midpoint, and *S* is the slope factor, inversely related to curve steepness near activation threshold.

*Activation time course of type II HCs.* We fit current traces using a custom MATLAB function (fitkin.m, *Source code 1*). For type II HCs lacking fast inactivation, outward current activation was fit with *Equation 2*.

$$I\left(t\right) = I_{SS} * \left(1 - exp(-\frac{t}{\tau_w})\right)^n + I_0 \tag{2}$$

$I_{SS}$ is steady-state current; $\tau_w$ is activation time constant (referred to elsewhere as $\tau_{Act}$); *n* is the state factor related to the number of closed states (typically constrained to 3); and $I_o$ is baseline current.

To measure activation and inactivation time course of $g_A$, we used *Equation 3* to fit outward K⁺ currents evoked by steps from –125 mV to above –50 mV (*Rothman and Manis, 2003*).

$$I\left(t\right) = I_{max} * \left(1 - exp(-\frac{t}{\tau_w})\right)^n * \left[1 - Z * \left(f * \left(1 - exp(-\frac{t}{\tau_{zf}})\right) + (1 - f) * \left(1 - exp(-\frac{t}{\tau_{zs}})\right)\right)\right] + I_0 \tag{3}$$

*Z* is total steady-state inactivation (0 ≤ Z < 1 means incomplete inactivation, which allows the equation to fit non-inactivating delayed rectifier currents); *f* is the fraction of fast inactivation relative to total inactivation; $I_{max}$ is maximal current; $\tau_{zf}$ (referred to elsewhere as $\tau_{Inact,Fast}$) and $\tau_{zs}$ are the fast and slow inactivation time constants. We chose to compare fit parameters at 30 ± 2 mV (91), where fast and slow inactivation were consistently separable and $g_A$ was maximized. In most *Kcna10⁻/⁻* and some striolar *Kcna10⁺/⁺,⁺/⁻* cells, where fast inactivation was absent and adjusted $R^2$ did not improve on a single-exponential fit by >0.01, we constrained *f* in *Equation 3* to 0 to avoid overfitting.

For *Peak G–V* relations, peak conductance was taken from fitted curves (*Equations 2 and 3*). To construct '*Steady-state*' *G–V* relations, we used current at 200 ms, which was only 6 ± 1% (94) greater than steady-state estimated from fits to *Equation 3* (*Figure 3C, D*).

Percent inactivation was calculated at 30 mV with *Equation 4*:

$$\% \, Inactivation = \left(I_{Peak} - I_{SS}\right) / I_{Peak} \tag{4}$$

$I_{Peak}$ is maximal current, and $I_{SS}$ is current at the end of a 200-ms voltage step.

The electrical resonance of type II HCs was quantified by fitting voltage responses to current injection steps (*Songer and Eatock, 2013*). We fit *Equation 5*, a damped sinusoid, to the voltage trace from half-maximum of the initial depolarizing peak until the end of the current step.

$$V(t) = V_{ss} + V_p * exp(-\frac{t}{\tau_e}) * \sin\left(2\pi f_e - \theta\right)$$ (5)

$V_{ss}$ is steady-state voltage; $V_p$ is the voltage of the peak response; $\tau_e$ is the decay time constant; $f_e$ is the fundamental frequency; and $\theta$ is the phase angle shift.

Quality factor, $Q_e$, was calculated with *Equation 6* (*Crawford and Fettiplace, 1981*).

$$Q_e = \left[\left(\pi f_e \tau_e\right)^2 + 0.25\right]^{1/2}$$ (6)

## Statistics

We give means ± SEM for normally distributed data, and otherwise, median and range. Data normality was assessed with the Shapiro–Wilk test for $n < 50$ and the Kolmogorov–Smirnov test for $n > 50$. To assess homogeneity of variance we used Levene's test. With homogeneous variance, we used two-way ANOVA for genotype and zone with the post hoc Tukey's test. When variance was non-homogeneous, we used one-way Welch ANOVA with the posthoc Games–Howell test. For data that were not normally distributed, we used the non-parametric one-way Kruskal–Wallis ANOVA (KWA) with posthoc Dunn's test. Effect size is Hedge's g (g). For age dependence, we used partial correlation coefficients controlling for genotype and zone. Statistical groups may have different median ages, but all have overlapping age ranges. In figures, asterisks represent p-value ranges as follows: *$p < 0.05$; **$p < 0.01$; ***$p < 0.001$; ****$p < 0.0001$.

## Immunohistochemistry

Mice were anesthetized with Nembutal (80 mg/kg), then perfused transcardially with 40 ml of physiological saline containing heparin (400 IU), followed by 2 ml/g body weight fixative (4% paraformaldehyde, 1% picric acid, and 5% sucrose in 0.1 M phosphate buffer at pH 7.4, sometimes with 1% acrolein). Vestibular epithelia were dissected in phosphate buffer, and tissues were cryoprotected in 30% sucrose-phosphate buffer overnight at 4°C. Otoconia were dissolved with Cal-Ex (Fisher Scientific) for 10 min. Frozen sections (35 μm) were cut with a sliding microtome. Immunohistochemistry was performed on free-floating sections. Tissues were first permeabilized with 4% Triton X-100 in phosphate-buffered saline (PBS) for 1 hr at room temperature, then incubated with 0.5% Triton X-100 in a blocking solution of 0.5% fish gelatin and 1% bovine serum albumin for 1 hr at room temperature. Sections were incubated with two to three primary antibodies for 72 hr at 4°C and with two to three secondary antibodies. Sections were rinsed with PBS between and after incubations and mounted on slides in Mowiol (Calbiochem).

## Acknowledgements

This study was supported by NIH grant R01 DC012347 to RAE and AL and an NSF Graduate Research Fellowship to HRM. We thank Drs. Thomas Friedman and Sherri Jones for the generous gift of the *Kcna10*[-/-] mouse line, and Drs. Zheng-Yi Chen and Deborah I Scheffer for bringing the expression of this subunit in mouse vestibular hair cells to our attention. We acknowledge Dr. Vicente Lumbreras for insights from his prior experiments on $g_A$ in mouse utricular hair cells, and thank him for helpful further discussions. We acknowledge Steven D Price for his help with immunocytochemistry. We thank Drs. Rebecca Lim and Ebenezer Yamoah for their critical feedback on the manuscript, and Drs. Rob Raphael and Aravind Chenrayan Govindaraju for feedback and many helpful discussions. We thank Drs. Joe Burns, Gabi Pregernig, and Lars Becker (Decibel Therapeutics, Inc) for helpful discussions.

## Additional information

### Funding

| Funder | Grant reference number | Author |
|---|---|---|
| National Institute on Deafness and Other Communication Disorders | R01 DC012347 | Anna Lysakowski Ruth Anne Eatock |
| National Science Foundation | Graduate Research Fellowship Program | Hannah R Martin |

The funders had no role in study design, data collection, and interpretation, or the decision to submit the work for publication.

### Author contributions

Hannah R Martin, Conceptualization, Data curation, Software, Formal analysis, Investigation, Visualization, Methodology, Writing - original draft, Project administration, Writing – review and editing; Anna Lysakowski, Funding acquisition, Investigation, Methodology, Project administration, Writing – review and editing; Ruth Anne Eatock, Conceptualization, Supervision, Funding acquisition, Validation, Visualization, Methodology, Writing - original draft, Project administration, Writing – review and editing

### Author ORCIDs

Hannah R Martin http://orcid.org/0000-0003-2028-5798
Anna Lysakowski https://orcid.org/0000-0001-6259-0294
Ruth Anne Eatock https://orcid.org/0000-0001-7547-2051

### Ethics

All procedures for handling animals followed the NIH Guide for the Care and Use of Laboratory Animals and were approved by the Institutional Animal Care and Use Committees of the University of Chicago (Animal Care and Use Procedure #72360) and the Office of Animal Care and Institutional Biosafety at the University of Illinois Chicago (Protocol for Animal Use #17106).

Reviewer #1 (Public review): https://doi.org/10.7554/eLife.94342.4.sa1
Reviewer #2 (Public review): https://doi.org/10.7554/eLife.94342.4.sa2
Reviewer #3 (Public review): https://doi.org/10.7554/eLife.94342.4.sa3
Author response https://doi.org/10.7554/eLife.94342.4.sa4

## Additional files

### Supplementary files

• Supplementary file 1. This file contains descriptive and comparative statistics on three additional analyses of the dataset. (**a**) Test of sex differences in hair cell $K_V$ channel data. (**b**) We did not detect a genotype effect on soma size of type I hair cells (HCs). (**c**) $I_{Kir}$ and $I_H$ were greater in the extrastriola (ES) than striola (S), but did not vary by genotype.

• MDAR checklist

• Source code 1. Fitkin.m is a MATLAB function to fit the activation and inactivation kinetics of $K_V$ currents to *Equations 2; 3*. Fitkin.m processed current–time traces and output parameters found in *Figure 3*, *Figure 3—figure supplement 1*, and *Table 3*.

• Source code 2. Fitzmann.m is a MATLAB function to fit the activation voltage dependence of $K_V$ currents to *Equation 1*. Fitzmann.m processed *G–V* data points and output parameters found in *Figure 1*, *Figure 3*, *Figure 7*, *Figure 1—figure supplement 1*, *Figure 3—figure supplement 2*, *Table 1*, *Table 4*, and *Supplementary file 1a*.

### Data availability

Data generated and analyzed in this study are available on Dryad (https://doi.org/10.5061/dryad.37pvmcvrw). Dryad hosts downloadable spreadsheets that are organized as follows: F1_sourcedata, numerical data for Figure 1; F1-fs1_sourcedata, numerical data for Figure 1-figure supplement 1;

F2_sourcedata, numerical data for Figure 2; F3_sourcedata, numerical data for Figure 3; F3-fs1_sourcedata, numerical data for Figure 3—figure supplement 1; F3-fs2_sourcedata, numerical data for Figure 3—figure supplement 2; F3-fs3_sourcedata, numerical data for Figure 3—figure supplement 3; F4_sourcedata, numerical data for Figure 4; F6_sourcedata, numerical data for Figure 6; F7_sourcedata, numerical data for Figure 7; F7-fs1_sourcedata, numerical data for Figure 7—figure supplement 1; F7-fs2_sourcedata, numerical data for Figure 7—figure supplement 2.

The following dataset was generated:

| Author(s) | Year | Dataset title | Dataset URL | Database and Identifier |
|---|---|---|---|---|
| Martin HR, Lysakowski A, Eatock RA | 2024 | The potassium channel subunit Kv1.8 (Kcna10) is essential for the distinctive outwardly rectifying conductances of type I and II vestibular hair cells | https://doi.org/10.5061/dryad.37pvmcvrw | Dryad Digital Repository, 10.5061/dryad.37pvmcvrw |

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
