## [Editor Report · eLife assessment]

This study provides direct evidence showing that K_V_1.8 channels provide the basis for several potassium currents in the two types of sensory hair cells found in the mouse vestibular system. This is an **important** finding because the nature of the channels underpinning the unusual potassium conductance g_K,L_ in type I hair cells has been under scrutiny for many years. The experimental evidence is **compelling** and the analysis is rigorous. The study will be of interest to cell and molecular biologists as well as vestibular and auditory neuroscientists.

---

## [Referee Report · Reviewer #1 (Public review)]

Summary:

In this paper the authors provide a thorough demonstration of the role that one particular type of voltage-gated potassium channel, Kv1.8, plays in a low voltage activated conductance found in type I vestibular hair cells. Along the way, they find that this same channel protein appears to function in type II vestibular hair cells as well, contributing to other macroscopic conductances. Overall, Kv1.8 may provide especially low input resistance and short time constants to facilitate encoding of more rapid head movements in animals that have necks. Combination with other channel proteins, in different ratios, may contribute to the diversified excitability of vestibular hair cells.

Strengths:

The experiments are comprehensive and clearly described, both in text and in the figures. Statistical analyses are provided throughout.

Weaknesses:

None.

---

## [Referee Report · Reviewer #2 (Public review)]

The focus of this manuscript was to investigate whether Kv1.8 channels, which have previously been suggested to be expressed in type I hair cells of the mammalian vestibular system, are responsible for the potassium conductance g_K,L_. This is an important study because g_K,L_ is known to be crucial for the function of type I hair cells, but the channel identity has been a matter of debate for the past 20 years. The authors have addressed this research topic by primarily investigating the electrophysiological properties of the vestibular hair cells from Kv1.8 knockout mice. Interestingly, g_K,L_ was completely abolished in Kv1.8-deficient mice, in agreement with the hypothesis put forward by the authors based on the literature. The surprising observation was that in the absence of Kv1.8 potassium channels, the outward potassium current in type II hair cells was also largely reduced. Type II hair cells express the largely inactivating potassium conductance g_K,A_, but not g_K,L_. The authors concluded that heteromultimerization of non-inactivating Kv1.8 and the inactivating Kv1.4 subunits could be responsible for the inactivating g_K,A_. Overall, the manuscript is very well written and most of the conclusions are supported by the experimental work. The figures are well described, and the statistical analysis is robust.

---

## [Referee Report · Reviewer #3 (Public review)]

Summary:

This paper by Martin et al. describes the contribution of a Kv channel subunit (Kv1.8, KCNA10) to voltage-dependent K+ conductances and membrane properties of type I and type II hair cells of the mouse utricle. Previous work has documented striking differences in K+ conductances between vestibular hair cell types. In particular amniote type I hair cells are known to express a non-typical low-voltage-activated K+ conductance (G_K,L_) whose molecular identity has been elusive. K+ conductances in hair cells from 3 different mouse genotypes (wildtype, Kv1.8 homozygous knockouts and heterozygotes) are examined here and whole cell patch-clamp recordings indicate a prominent role for Kv1.8 subunits in generating G_K,L_. Results also interestingly support a role for Kv1.8 subunits in type II hair cell K+ conductances; inactivating conductances in null mice are reduced in type II hair cells from striola and extrastriola regions of the utricle. Kv1.8 is therefore proposed to contribute as a pore-forming subunit for 3 different K+ conductances in vestibular hair cells. The impact of these conductances on membrane responses to current steps is studied in current clamp. Pharmacological experiments use XE991 to block some residual Kv7-mediated current in both hair cell types, but no other pharmacological blockers are used. In addition immunostaining data are presented and raise some questions about Kv7 and Kv1.8 channel localization. Overall, the data present compelling evidence that removal of Kv1.8 produces profound changes in hair cell membrane conductances and sensory capabilities. These changes at hair cell level suggest vestibular function would be compromised and further assessment in terms of balance behavior in the different mice would be interesting.

Strengths:

This study provides strong evidence that Kv1.8 subunits are major contributors to the unusual K+ conductance in type I hair cells of the utricle. It also indicates that Kv1.8 subunits are important for type II hair cell K+ conductances because Kv1.8-/- mice lacked an inactivating A conductance and had reduced delayed rectifier conductance compared to controls. A comprehensive and careful analysis of biophysical profiles is presented of expressed K+ conductances in 3 different mouse genotypes. Voltage-dependent K+ currents are rigorously characterized at a range of different ages and their impact on membrane voltage responses to current input is studied. Some pharmacological experiments are performed in addition to immunostaining to bolster the conclusions from the biophysical studies. The paper has a significant impact in showing the role of Kv1.8 in determining utricular hair cell electrophysiological phenotypes.

Weaknesses:

(1) From previous work it is known that G_K,L_ in type I hair cells has unusual ion permeation and pharmacological properties that differ greatly from type II hair cell conductances. Notably G_K,L_ is highly permeable to Cs+ as well as K+ ions and is slightly permeable to Na+. It is blocked by 4-aminopyridine and divalent cations (Ba2+, Ca2+, Ni2+), enhanced by external K+ and modulated by cyclic GMP. The question arises-if Kv1.8 is a major player and pore-forming subunit in type I and type II cells (and cochlear inner hair cells as shown by Dierich et al. 2020) how are subunits modified to produce channels with very different properties? A role for Kv1.4 channels (gA) is proposed in type II hair cells based on previous findings in bird hair cells. However, hair cell specific partner interactions with Kv1.8 that result in GK, L in type I hair cells and Cs+ impermeable, inactivating currents in type II hair cells remain for the most part unexplored.

(2) Data from patch-clamp and immunocytochemistry experiments are not in close alignment. XE991 (Kv7 channel blocker) decreases remaining K+ conductance in type I and type II hair cells from null mice supporting the presence of Kv7 channels in hair cells (Fig. 7). Also, Holt et al. (2007) previously showed inhibition of G_K,L_ in type I hair cells (but not delayed rectifier conductance in type II hair cells) using a dominant negative construct of Kv7.4 channels. However, immunolabelling indicates Kv7.4 channels on the inner face of calyx terminals adjacent to hair cells (Fig. 5). Some reconciliation of these findings is needed.

(3) A previous paper reported that a vestibular evoked potential was abnormal in Kv1.8-/- mice (Lee et al. 2013) as briefly mentioned (lines 94-95). It would be really interesting to know if any vestibular-associated behaviors and/or hearing loss were observed in the mice populations. If responses are compromised at the sensory hair cell level across different zones, degradation of balance function would be anticipated and should be elucidated.

---

## [Author Response]

The following is the authors’ response to the previous reviews.

**Recommendations for the authors:**

**Reviewer #1 (Recommendations For The Authors):**
Line 127. Provide a few more words describing the voltage protocol. To the uninitiated, panels A and B will be difficult to understand. "The large negative step is used to first close all channels, then probe the activation function with a series of depolarizing steps to re-open them and obtain the max conductance from the peak tail current at -36 mV. "

We have revised the text as suggested (revision lines 127 to Line 131): “From a holding potential within the g_K,L_ activation range (here –74 mV), the cell is hyperpolarized to –124 mV, negative to EK and the activation range, producing a large inward current through open g_K,L_ channels that rapidly decays as the channels deactivate. We use the large transient inward current as a hallmark of g_K,L_. The hyperpolarization closes all channels, and then the activation function is probed with a series of depolarizing steps, obtaining the max conductance from the peak tail current at –44 mV (Fig. 1A).”

Incidentally, why does the peak tail current decay?

We added this text to the figure legend to explain this: “For steps positive to the midpoint voltage, tail currents are very large. As a result, K^+^ accumulation in the calyceal cleft reduces driving force on K^+^, causing currents to decay rapidly, as seen in A (Lim et al., 2011).”

The decay of the peak tail current is a feature of g_K,L_ (large K^+^ conductance) and the large enclosed synaptic cleft (which concentrates K^+^ that effluxes from the HC). See Govindaraju et al. (2023) and Lim et al. (2011) for modeling and experiments around this phenomenon.

Line 217-218. For some reason, I stumbled over this wording. Perhaps rearrange as "In type II HCs absence of Kv1.8 significantly increased R_in_ and tau_RC_. There was no effect on V_rest_ because the conductances to which Kv1.8 contributes, g_A_ and g_DR_ activate positive to the resting potential. (so which K conductances establish V_rest_???).

We kept our original wording because we wanted to discuss the baseline (V_rest_) before describing responses to current injection.

->V_rest_ is presumably maintained by ATP-dependent Na/K exchangers (ATP1a1), HCN, Kir, and mechanotransduction currents. Repolarization is achieved by delayed rectifier and A-type K^+^ conductances in type II HCs.

Figure 4, panel C - provides absolute membrane potential for voltage responses. Presumably, these were the most 'ringy' responses. Were they obtained at similar Vm in all cells (i.e., comparisons of Q values in lines 229-230).

We added the absolute membrane potential scale. Type II HC protocols all started with 0 pA current injection at baseline, so they were at their natural V_rest_, which did not differ by genotype or zone. Consistent with Q depending on expression of conductances that activate positive to V_rest_, Q did not co-vary with V_rest_ (Pearson’s correlation coefficient = 0.08, p = 0.47, n = 85).

Lines 254. Staining is non-specific? Rather than non-selective?

Yes, thanks - Corrected (Line 264).

Figure 6. Do you have a negative control image for Kv1.4 immuno? Is it surprising that this label is all over the cell, but Kv1.8 is restricted to the synaptic pole?

We don’t have a null-animal control because this immunoreactivity was done in rat. While the cuticular plate staining was most likely nonspecific because we see that with many different antibodies, it’s harder to judge the background staining in the hair cell body layer. After feedback from the reviewers, we decided to pull the K_V_1.4 immunostaining from the paper because of the lack of null control, high background, and inability to reproduce these results in mouse tissue. In our hands, in mouse tissue, both mouse and rabbit anti-K_V_1.4 antibodies failed to localize to the hair cell membrane. Further optimization or another method could improve that, but for now the single-cell expression data (McInturff et al., 2018) remain the strongest evidence for K_V_1.4 expression in murine type II hair cells.

Lines 400-404. Whew, this is pretty cryptic. Expand a bit?

We simplified this paragraph (revision lines 411-413): “We speculate that g_A_ and g_DR_(K_V_1.8) have different subunit composition: g_A_ may include heteromers of K_V_1.8 with other subunits that confer rapid inactivation, while g_DR_(K_V_1.8) may comprise homomeric K_V_1.8 channels, given that they do not have N-type inactivation .”

Line 428. 'importantly different ion channels'. I think I understand what is meant but perhaps say a bit more.

Revised (Line 438): “biophysically distinct and functionally different ion channels”.

Random thought. In addition to impacting R_in_ and Tau_RC,_ do you think the more negative V_rest_ might also provide a selective advantage by increasing the driving force on K entry from endolymph?

When the calyx is perfectly intact, g_K,L_ is predicted to make V_rest_ less negative than the values we report in our paper, where we have disturbed the calyx to access the hair cell (–80, Govindaraju et al., 2023, *vs.* –87 mV, here). By enhancing K^+^ accumulation in the calyceal cleft, the intact calyx shifts E_K_—and V_rest_—positively (Lim et al., 2011), so the effect on driving force may not be as drastic as what you are thinking.

**Reviewer #2 (Recommendations For The Authors):**
**(**1) Introduction: wouldn't the small initial paragraph stating the main conclusion of the study fit better at the end of the background section, instead of at the beginning?

Thank you for this idea, we have tried that and settled on this direct approach to let people know in advance what the goals of the paper are.

(2) Pg.4: The following sentence is rather confusing "Between P5 and P10, we detected no evidence of a non-g_K,L_ K_V_1.8-dependent.....". Also, Suppl. Fig 1A seems to show that between P5 and P10 hair cells can display a potassium current having either a hyperpolarised or depolarised V_half_. Thus, I am not sure I understand the above statement.

Thank you for pointing out unclear wording. We used the more common “delayed rectifier” term in our revision (Lines 144-147): “Between P5 and P10, some type I HCs have not yet acquired the physiologically defined conductance, g_K,L_.. N effects of K_V_1.8 deletion were seen in the delayed rectifier currents of immature type I HCs (Suppl. Fig. 1B), showing that they are not immature forms of the K_V_1.8-dependent g_K,L_ channels. ”

(3) For the reduced Cm of hair cells from Kv1.8 knockout mice, could another reason be simply the immature state of the hair cells (i.e. lack of normal growth), rather than less channels in the membrane?

There were no other signs to suggest immaturity or abnormal growth in K_V_1.8–/– hair cells or mice. Importantly, type II HCs did not show the same C_m_ effect.

We further discussed the capacitance effect in lines 160-167: “C_m_ scales with surface area, but soma sizes were unchanged by deletion of K_V_1.8 (Suppl. Table 2). Instead, C_m_ may be higher in K_V_1.8+/+ cells because of g_K,L_ for two reasons. First, highly expressed trans-membrane proteins (see discussion of g_K,L_ channel density in Chen and Eatock, 2000) can affect membrane thickness (Mitra et al., 2004), which is inversely proportional to specific C_m_. Second, g_K,L_ could contaminate estimations of capacitive current, which is calculated from the decay time constant of transient current evoked by small voltage steps *outside* the operating range of any ion channels. g_K,L_ has such a negative operating range that, even for Vm negative to –90 mV, some g_K,L_ channels are voltage-sensitive and could add to capacitive current.”

(4) Methods: The electrophysiological part states that "For most recordings, we used .....". However, it is not clear what has been used for the other recordings.

Thanks for catching this error, a holdover from an earlier ms. version. We have deleted “For most recordings” (revision line 466).

Also, please provide the sign for the calculated 4 mV liquid junction potential.

Done (revision line 476).

**Reviewer #3 (Recommendations For The Authors):**
**(**1) Some of the data in panels in Fig. 1 are hard to match up. The voltage protocols shown in A and B show steps from hyperpolarized values to -71mV (A) and -32 mV (B). However, the value from A doesn't seem to correspond with the activation curve in C.

Thank you for catching this. We accidentally showed the control I-X curve from a different cell than that in A. We now show the G-V relation for the cell in A.

Also the V_half_ in D for -/- animals is ~-38 mV, which is similar to the most positive step shown in the protocol.

The most positive step in Figure 1B is actually –25 mV. The uneven tick labels might have been confusing, so we re-labeled them to be more conventional.

Were type I cells stepped to more positive potentials to test for the presence of voltage-activated currents at greater depolarizations? This is needed to support the statement on lines 147-148.

We added “no additional K+ conductance activated up to +40 mV” (revision line 149-150). Our standard voltage-clamp protocol iterates up to ~+40 mV in K_V_1.8–/– hair cells, but in Figure 1 we only showed steps up to –25 mV because K^+^ accumulation in the synaptic cleft with the calyx distorts the current waveform even for the small residual conductances of the knockouts. K_V_1.8–/– hair cells have a main KV conductance with a V_half_ of ~–38 mV, as shown in Figure 1, and we did not see an additional K_V_ conductance that activated with a more positive V_half_ up to +40 mV.

(2) Line 151 states "While the cells of Kv1.8-/- appeared healthy..." how were epithelia assessed for health? Hair cells arise from support cells and it would be interesting to know if Kv1.8 absence influences supporting cells or neurons.

We added our criteria for cell health to lines 477-479: “K_V_1.8–/– hair cells appeared healthy in that cells had resting potentials negative to –50 mV, cells lasted a long time (20-30 minutes) in ruptured patch recordings, membranes were not fragile, and extensive blebbing was not seen.”

Supporting cells were not routinely investigated. We characterized calyx electrical activity (passive membrane properties, voltage-gated currents, firing pattern) and didn’t detect differences between +/+, +/–, and –/– recordings (data not shown). K_V_1.8 was not detected in neural tissue (Lee et al., 2013).

(3) Several different K+ channel subtypes were found to contribute to inner hair cell K+ conductances (Dierich et al. 2020) but few additional K+ channel subtypes are considered here in vestibular hair cells. Further comments on calcium-activated conductances (lines 310-317) would be helpful since apamin-sensitive SK conductances are reported in type II hair cells (Poppi et al. 2018) and large iberiotoxin-sensitive BK conductances in type I hair cells (Contini et al. 2020). Were iberiotoxin effects studied at a range of voltages and might calcium-dependent conductances contribute to the enhanced resonance responses shown in Fig. 4?

We refer you to lines 310-317 in the original ms (lines 322-329 in the revised ms), where we explain possible reasons for not observing I_K_(Ca) in this study.

(4) Similar to G_K,L_ erg (Kv11) channels show significant Cs+-permeability. Were experiments using Cs+ and/or Kv11 antagonists performed to test for Kv11?

No. Hurley et al. (2006) used Kv11 antagonists to reveal Kv11 currents in rat utricular type I hair cells with perforated patch, which were also detected in rats with single-cell RT-PCR (Hurley et al. 2006) and in mice with single-cell RNAseq (McInturff et al., 2018). They likely contribute to hair cell currents, alongside Kv7, Kv1.8, HCN1, and Kir.

(5) Mechanosensitive ("MET") channels in hair cells are mentioned on lines 234 and 472 (towards the end of the Discussion), but a sentence or two describing the sensory function of hair cells in terms of MET channels and K+ fluxes would help in the Introduction too.

Following this suggestion we have expanded the introduction with the following lines (78-87): “Hair cells are known for their large outwardly rectifying K+ conductances, which repolarize membrane voltage following a mechanically evoked perturbation and in some cases contribute to sharp electrical tuning of the hair cell membrane. Because g_K,L_ is unusually large and unusually negatively activated, it strongly attenuates and speeds up the receptor potentials of type I HCs (Correia et al., 1996; Rüsch and Eatock, 1996b). In addition, g_K,L_ augments a novel non-quantal transmission from type I hair cell to afferent calyx by providing open channels for K^+^ flow into the synaptic cleft (Contini et al., 2012, 2017, 2020; Govindaraju et al., 2023), increasing the speed and linearity of the transmitted signal (Songer and Eatock, 2013).”

(6) Lines 258-260 state that GKL does not inactivate, but previous literature has documented a slow type of inactivation in mouse crista and utricle type I hair cells (Lim et al. 2011, Rusch and Eatock 1996) which should be considered.

Lim et al. (2011) concluded that K^+^ accumulation in the synaptic cleft can explain much of the apparent inactivation of g_K,L_. In our paper, we were referring to fast, N-type inactivation. We changed that line to be more specific; new revision lines 269-271: “K_V_1.8, like most K_V_1 subunits, does not show fast inactivation as a heterologously expressed homomer (Lang et al., 2000; Ranjan et al., 2019; Dierich et al., 2020), nor do the K_V_1.8-dependent channels in type I HCs, as we show, and in cochlear inner hair cells (Dierich et al., 2020).”

(7) Lines 320-321 Zonal differences in inward rectifier conductances were reported previously in bird hair cells (Masetto and Correia 1997) and should be referenced here.

Zonal differences were reported by Masetto and Correia for type II but not type I avian hair cells, which is why we emphasize that we found a zonal difference in I-H in type I hair cells. We added two citations to direct readers to type II hair cell results (lines 333-334): “The g_K,L_ knockout allowed identification of zonal differences in I_H_ and I_Kir_ in type I HCs, previously examined in type II HCs (Masetto and Correia, 1997; Levin and Holt, 2012).”

Also, Horwitz et al. (2011) showed HCN channels in utricles are needed for normal balance function, so please include this reference (see line 171).

Done (line 184).

(8) Fig 6A. Shows Kv1.4 staining in rat utricle but procedures for rat experiments are not described. These should be added. Also, indicate striola or extrastriola regions (if known).

We removed K_V_1.4 immunostaining from the paper, see above.

(9) Table 6, ZD7288 is listed -was this reagent used in experiments to block G_h_? If not please omit.

ZD7288 was used to block g_H_ to produce a clean h-infinity curve in Figure 6, which is described in the legend.

(10) In supplementary Fig. 5A make clear if the currents are from XE991 subtraction. Also, is the G-V data for single cell or multiple cells in B? It appears to be from 1 cell but ages P11-505 are given in legend.

The G-V curve in B is from XE991 subtraction, and average parameters in the figure caption are for all the K_V_1.8–/– striolar type I hair cells where we observed this double Boltzmann tail G-V curve. I added detail to the figure caption to explain this better.

(11) Supplementary Fig. 6A claims a fast activation of inward rectifier K+ channels in type II but not type I cells-not clear what exactly is measured here.

We use “fast inward rectifier” to indicate the inward current that increases within the first 20 ms after hyperpolarization from rest (I_Kir_, characterized in Levin & Holt, 2012) in contrast to HCN channels, which open over ~100 ms. We added panel C to show that the activation of I_Kir_ is visible in type II hair cells but not in the knockout type I hair cells that lack g_K,L_. I_Kir_ was a reliable cue to distinguish type I and type II hair cells in the knockout.

For our actual measurements in Fig 6B, we quantified the current flowing after 250 ms at –124 mV because we did not pharmacologically separate I_Kir_ and I_H_.

Could the XE991-sensitive current be activated and contributing?

The XE991-sensitive current could decay (rapidly) at the onset of the hyperpolarizing step, but was not contributing to our measurement of I_Kir_­ and I_H_, made after 250 ms at –124 mV, at which point any low-voltage-activated (LVA) outward rectifiers have deactivated. Additionally, the LVA XE991-sensitive currents were rare (only detected in some striolar type I hair cells) and when present did not compete with fast I_Kir_, which is only found in type II hair cells.

Also, did the inward rectifier conductances sustain any outward conductance at more depolarized voltage steps?

For the K_V_1.8-null mice specifically, we cannot answer the question because we did not use specific blocking agents for inward rectifiers. However, we expect that there would only be sustained outward IR currents at voltages between E_K_ and ~-60 mV: the foot of I_Kir_’s I-V relation according to published data from mouse utricular hair cells – e.g., Holt and Eatock 1995, Rusch and Eatock 1996, Rusch et al. 1998, Horwitz et al., 2011, etc. Thus, any such current would be unlikely to contaminate the residual outward rectifiers in Kv1.8-null animals, which activate positive to ~-60 mV.

(I-HCN is also not a problem, because it could only be outward positive to its reversal potential at ~-40 mV, which is significantly positive to its voltage activation range.)